# Channel Selection of Closed-Loop Supply Chain for Scrapped Agricultural Machines Remanufacturing

**Linan Zhou [1], Gengui Zhou [2,\*], Hangying Li [3] and Jian Cao [2]**

1   Business School, Hangzhou City University, Hangzhou 310015, China; zhoulinan@zucc.edu.cn
2   College of Management, Zhejiang University of Technology, Hangzhou 310023, China
3   Zhejiang University Press, Zhejiang University, Hangzhou 310007, China
\*   Correspondence: ggzhou@zjut.edu.cn

**Abstract:** Along with economic development and technological innovation, rapid expansion of agricultural machinery has sparked widespread concern. In particular, the superiority of scrapped agricultural machinery recycling and remanufacturing in improving environmental sustainability, economic benefits, and carbon emission reduction has garnered public attention. Based on this reality, this study constructs models for five different agricultural machinery recycling channels according to different actors involved in recovery, dismantling, and remanufacturing. Each model's equilibrium is determined by mathematical deduction. The applicable condition of each model and the influence of multiple factors are analyzed. The results indicate the following: (i) no single recycling channel is definitely superior to others, and different channels have their own applicable conditions that are influenced by transfer payments, supply chain cooperation, recycling prices, and supply and demand; (ii) cooperative scenarios are more conducive to attracting third-party enterprises to participate to increase supply chain revenue; (iii) rise in recovery and remanufacturing prices may lead to divergence among supply chain parties on channel selection; and (iv) oversupply requires government subsidies to maintain recycling and remanufacturing.

**Keywords:** agricultural machinery remanufacturing; closed-loop supply chain; channel selection

## 1. Introduction

As an important tool in agricultural production, agricultural machinery is critical to agricultural modernization. Widespread use of agricultural machinery is a prerequisite for promoting development of agricultural mechanization. China's agricultural machinery development has progressed through three stages: the initial stage, the sustainable development stage of small agricultural machinery, and the rapid development stage of large electric agricultural machinery [1]. To better adapt to the development requirements of modern agriculture and improve agricultural production's technical level and mechanization process, China has successively promulgated the Promotion Law of Agricultural Mechanization and the Opinions of the State Council on Promoting the Sound and Rapid Development of Agricultural Mechanization and Agricultural Machinery Industry. In recent years, social ownership of agricultural machinery products in China has increased significantly owing to relevant national agricultural machinery policies [2].

With the rapid increase in agricultural machinery, the number of agricultural machines that need to be scrapped is also growing quickly. There are two main reasons for scrapping agricultural machinery [3]. One is that some agricultural machinery is seriously "out of service" and its continued use will result in serious pollution or environmental injury. It should be scrapped in the interests of environmental protection and safe production. Second, in the long-term operating environment, some agricultural machines' fuselage and core components are worn and consumed greatly, causing them to lose the ability to engage in agricultural production. They should be recycled. As with other scrapped

products, scrapped agricultural machines contain huge economic value [4]. Realizing recycling of scrapped agricultural machines is akin to creating a "green oilfield", with enormous potential.

Many agricultural machinery products currently being scrapped in China have not been recycled or reused on time, resulting in some relatively negative effects [5]. Some agricultural machines that should be scrapped are still used in the field. These agricultural machines have low efficiency, insufficient power, high energy consumption, and serious pollution [6]. A portion of the agricultural machinery that needs to be scrapped flows into underground channels, and, after being refitted by illegal sellers, it flows into the market, posing significant safety risks to consumers who buy and use it [5]. Therefore, effective recovery management of waste agricultural machinery needs to be strengthened. Accelerating recycling is conducive to environmental protection and can result in recycling and sustainable resource utilization.

In light of this, the Chinese government issued guidance in 2012 on implementing pilot subsidies for scrapping and updating agricultural machinery. The pilot program for scrapping and updating agricultural machinery was carried out in 11 provinces: Shanxi, Jiangsu, Zhejiang, Anhui, Shandong, Henan, Xinjiang, Ningbo, Qingdao, Xinjiang, and Heilongjiang. In 2015, the number of pilot provinces increased to 15, with Hebei, Jiangxi, Hubei, Hunan, Guangxi, Shaanxi, and Gansu newly added. In the pilot counties, subsidies are granted to farmers, herders, fishermen, and forest farm workers who scrap old machines and replace them with new ones in accordance with the law. Since the pilot program's inception ten years ago, all regions have actively explored and innovated, yielding remarkable results [5]. Approximately 500,000 old agricultural machines have been scrapped, and more than 20 large enterprises representing the entire industry have joined this program, which the government has announced will continue to expand [7].

The recycling enterprise shall disassemble the collected agricultural machinery for remanufacturing under the supervision of the county agricultural mechanization department [5]. The closed-loop supply chain (hereinafter CLSC) for remanufacturing scrapped agricultural machines primarily involves (i) agricultural machinery manufacturers, (ii) agricultural machinery sellers, (iii) third-party dismantling enterprises, and (iv) third-party remanufacturers, with the process primarily involving recovery, disassembly, cleaning, remanufacture, assembly, testing, and packaging [8,9]. Compared to the traditional manufacturing industry, the remanufacturing industry lacks a series of production links that cause serious environmental pollution, such as steel-making and casting, significantly reducing the environmental impact. With the Chinese government's strategic goal of reaching a carbon peak by 2030 and carbon neutrality by 2060, the remanufacturing industry has become increasingly prominent [10]. The Chinese government's policies and regulations (see Table 1) have fully demonstrated its determination and planning to promote the sustainable development of the agricultural machinery remanufacturing industry.

Driven by such policies and regulations, all the agricultural machinery manufacturers, sellers, dismantling enterprises, and remanufacturers have gradually begun to expand the scale of agricultural machinery remanufacturing [7]. However, the number of discarded agricultural machines is still only approximately 30,000 annually, which is far less than the number of old machines in rural areas. The main reason for this is that construction and selection of recycling and remanufacturing channels are still unclear, and relevant enterprises are not paying enough attention to them [5]. Each enterprise makes short-term decisions, resulting in chaotic and shifting channels, resulting in a serious obstacle to the desire of farmers to update agricultural machines and low profits obtained by enterprises through recycling and remanufacturing. Therefore, the CLSC for scrapped agricultural machines remanufacturing is difficult to develop continuously due to a lack of scientific decision-making strategies for channel selection. Thus, as the purpose of this study, investigating the recycling channels that comprise the aforementioned enterprises and providing members of the agricultural machinery remanufacturing supply chain with channel selection guidance are of practical significance to form a sustainable and effective

CLSC. The practice in China's agricultural machinery remanufacturing industry is highly motivating for this research.

**Table 1.** Relevant policies and regulations on agricultural machinery recycling and remanufacturing in China.

| Promulgation Time | Policies and Regulations | Main Purpose |
|---|---|---|
| 2004 | Law of the People's Republic of China on the Promotion of Agricultural Mechanization | Encourage and support farmers and agricultural production and operation organizations to use advanced and applicable agricultural machinery and promote agricultural mechanization. |
| 2008 | Prohibition and scrapping standards for tractors (GB/T 16877-2008) | Regulate the technical requirements and economic indicators for disabling and scrapping tractors. |
| 2010 | Technical conditions for disabling and scrapping combine harvesters (NY/T 1875-2010) | Regulate the technical requirements for disabling and scrapping the combine harvesters. |
| 2012 | The pilot program for scrapping and updating agricultural machinery | Regulate the subsidies for scrapping and updating agricultural machinery. |
| 2017 | The 13th Five Year Plan for the Development of National Agricultural Mechanization | Encourage and support: (i) Informatization of agricultural mechanization; (ii) Agricultural machinery safety; (iii) Agricultural machinery intelligence. |
| 2018 | Guiding Opinions of the State Council on Accelerating the Transformation and Upgrading of Agricultural Mechanization and Agricultural Machinery Equipment Industry | Accelerate the transformation of the agricultural machinery and equipment industry to high-quality development and promote the upgrading of agricultural mechanization. |
| 2018 | Notice on Guiding Opinions on the Implementation of Agricultural Machinery Purchase Subsidies from 2018 to 2020 | Support and guide the overall, high-quality, and efficient development of agricultural mechanization, promote the supply-side structural reform of agriculture, and help implement the rural revitalization strategy. |

Source of information: the Ministry of Agriculture and Rural Affairs of the People's Republic of China (http://www.njhs.moa.gov.cn/, accessed on 7 March 2023).

## 2. Literature Review

A CLSC is a network that includes returns processes, and the manufacturer has the intent of capturing additional value and further integrating all supply chain activities [11]. Therefore, a CLSC considers not only production and sale of products but also recovery, reuse, and remanufacturing of waste products, allowing for resource recovery and sustainable development [12,13]. Because CLSC research is critical for conserving resources and maximizing the value of goods, an increasing number of scholars have discussed CLSC management from various perspectives [14–18]. To our knowledge, existing works of literature have paid much attention to remanufacturing of mechanical products, such as automobiles, agricultural machines, and other similar machines, with selection of recycling channels in CLSC and differentiated pricing strategies for remanufactured products standing out as the two most-concerned topics at present.

### 2.1. Selection of Recycling Channels in CLSC

A recycling channel is a combination of recovery, dismantling, and remanufacturing processes, which has attracted the attention of many researchers. The first stream of

literature focuses on recovery channels, which involve who collects the used products. Specifically, the common recovery channels in a CLSC include: (i) direct recovery by the manufacturer, (ii) manufacturer entrusts retailer to recover, and (iii) manufacturer entrusts third-party enterprises to recover [19]. Among these modes, the retailer recovery mode appears to be dominant in the early stages [20]. For example, Savaskan et al. [21] investigated the problem of recovery channel selection between one manufacturer and two competitive retailers in remanufacturing reverse logistics. According to Choi et al. [22], a remanufacturing system's efficiency is highly related to a supply chain agent's proximity to the market; thus, the retailer-led channel provides the most effective CLSC. Furthermore, suppose recovery quantity is related to both retail price and recovery price: when the cost difference between new and remanufactured products is large, manufacturers tend to use the double-retailer recovery mode [23].

With the development of the reverse logistics industry, the third-party collectors that have developed rapidly are also of concern [24]. For instance, Giovanni and Zaccour [25] investigated whether manufacturers could recover on their own or outsource recovery to retailers or third-party enterprises; Feng et al. [26] expanded the research to include three scenarios: a single traditional recovery channel, a single online-recovery channel, and a hybrid dual-recovery channel. Based on the dual-recovery channel research, Huang and Wang [27] further analyzed how cost disruptions affect a manufacturer's channel choice. Meanwhile, Ranjbar et al. [28] proposed collection decisions under channel leadership and Zheng et al. [29] presented a supply chain coordinating mechanism based on the recovery channels. Furthermore, the impact of eco-design and CSR investment on recovery channel selection has also been studied [30,31]. Moreover, Matsui [32] investigated the best time for a collector to announce the recovery price.

The above studies are the mainstream of recycling channel research. Further, the second stream of literature focuses on remanufacturing channels: who produces the remanufactured products. Wang et al. [33] and Zhang et al. [34] discussed whether remanufacturing should be performed in-house or outsourced from the perspectives of manufacturers and retailers, respectively; Wang et al. [35] further analyzed manufacturers' recycling preference, supply chain controllability, and the impact on retailers and remanufacturers in a CLSC. Based on third-party collectors and third-party remanufacturers, Kushwaha et al. [36] considered the selection decisions of five different remanufacturing channels.

In summary, the existing research mainly focuses on selection of recovery channels. Some scholars have conducted research on selection of remanufacturing channels, but the current research on channel selection for the entire process of a CLSC, including recovery, dismantling, and remanufacturing, is still limited.

### 2.2. Differentiated Pricing Strategies for Remanufactured Products

According to Ferrer and Swaminathan [37], in the market, consumers usually have different willingness to pay for new and remanufactured products, so adopting differentiated pricing strategies is more realistic and worthy of study. Debo et al. [38] studied the pricing problem faced by a manufacturer considering introducing a remanufacturable product in a market with heterogeneous consumers, paving the way for further research in this field.

The first stream of the relevant literature focuses on the differentiated pricing strategies based on the manufacturer. Ferguson and Toktay [39] considered the potential profit loss due to external remanufacturing competition and analyzed differentiated pricing strategies to assist manufacturers; Based on study of manufacturer's pricing strategies in monopoly and duopoly environments [40], Ferrer and Swaminathan [37] investigated manufacturers' strategy for producing new and remanufactured products when the manufacturer has a monopoly on the markets; Chen and Chang [41] proposed a dynamic pricing strategy for new and remanufactured products that is dependent on the phases of product lifecycle in a CLSC. Meanwhile, Li et al. [42] found the optimal pricing strategy for remanufacturing when both the remanufacturing yield and demand for remanufactured products are ran-

dom. Moreover, Liu et al. [43] investigated the optimal production and pricing strategies for a monopolistic manufacturer engaged in remanufacturing.

The second stream of the relevant literature focuses on differentiated pricing strategies based on consumer demand. Essoussi and Linton [44] investigated the price premium that consumers are willing to pay for products with reused or recycled content; Ma et al. [45] studied a firm's optimal pricing decisions and presented the thresholds that determine whether the firm should offer "trade old for new" and "trade old for remanufactured" programs. Subsequently, Ma et al. [46] further considered consumers' double reference effects and studied the pricing strategies of manufacturers selling both new and remanufactured products; Zhu and Wang [47] investigated the conditions under which a trade-in program for remanufactured products should be implemented and the pricing strategy under the constraint of consumer participation. Hanh and Chen [48] discussed the selling prices for multiple differentiated versions of new and remanufactured products, assuming that the consumer demand is price- and reusability-dependent.

Other relevant literature focuses on differentiated pricing strategies based on supply chain partners and the external environment. Bulmuş et al. [49] simultaneously considered product acquisition management and pricing of the remanufactured products for hybrid manufacturing and remanufacturing systems. Meanwhile, Mitra [50] suggested that the combined profitability and market share of the (re)manufacturer on account of new and remanufactured product sales improve over new product sales only in a duopoly environment. Yenipazarli [51] investigated the impact of emissions taxes on the optimal production and pricing decisions of a manufacturer that could remanufacture its own product. Huang et al. [52] explored the optimal pricing decisions for a retailer-dominated CLSC with a triple recycling channel in the construction machinery remanufacturing industry. Last, Li et al. [53] compared the acquisition strategies of used products and the pricing strategies of new products and remanufactured products in different manufacturing–remanufacturing systems.

Based on the above, the existing research on differentiated pricing strategies is mainly based on manufacturers, consumer demand, supply chain partners, and the external environment, while the research on the pricing strategies of new and remanufactured products based on different recycling and remanufacturing channels is still limited.

*2.3. Research Gap*

Scholars have conducted extensive research on recycling channel selection and differentiated pricing strategies in CLSCs based on the information presented above. However, the current research primarily focuses on selection of recovery channels; research on channel selection for the entire process of a CLSC, including recovery, dismantling, and remanufacturing, is still limited. Since the structure of agricultural machinery is relatively simpler than that of large construction machinery and automobiles, the recovery, dismantling, and remanufacturing channels of agricultural machinery also involve more third-party enterprises, which is evident in China. Therefore, it is of great practical significance to study the problem of channel selection for the whole process of a CLSC. In addition, numerous studies have shown that differentiated pricing strategies should be adopted for new and remanufactured products, but previous studies are mainly based on manufacturers, consumer demand, supply chain partners, and the external environment. Research on the pricing strategies of new and remanufactured products based on different recycling and remanufacturing channels is still limited. Furthermore, research is lacking on the influence of differentiated pricing strategies on recycling channel selection in CLSCs.

Therefore, as an extension of relevant channel selection research [35,36] and pricing strategy research [49,53], this paper addresses both issues and aims to study the conditions that make selection of recycling channels more beneficial to members of CLSCs for agricultural machinery remanufacturing when there is a price difference between new and remanufactured products. The present study examined five agricultural machinery recycling and remanufacturing channels according to different actors involved in recovery,

dismantling, and remanufacturing and analyzed the pricing decisions of supply chain members. On this basis, numerical analysis is used to investigate the impact of various factors in different agricultural machinery recycling channels. Finally, we obtain the applicable conditions for CLSC channel selection by comparing the profits of all parties and the overall profit of the supply chain.

## 3. Model and Assumptions

### 3.1. Model Description

In real-word scenarios, agricultural machinery sellers or third-party dismantling enterprises recover scrapped agricultural machines, agricultural machinery manufacturers or dismantling enterprises dismantle the recovered products, and manufacturers or third-party remanufacturers handle the remanufacturing process [5,8,9]. Furthermore, due to issuance of a business license, only sellers are permitted to sell agricultural machinery, including new and remanufactured products. Independent recycling and remanufacturing of scrapped agricultural machines by manufacturers can achieve the optimization process of green design, green production, and green recycling, which is conducive to reducing resource consumption, improving the reuse rate of raw materials, and creating a good social image for manufacturers. However, this mode does not always form a scale effect, which can be compensated for by third-party enterprises. Hence, various recycling and remanufacturing models exist in today's agricultural machinery market [36].

According to our investigation, the existing agricultural machinery recycling and remanufacturing models are summarized in Table 2 based on the various actors involved in recovery, dismantling, and remanufacturing. The initials of each entity engaged in the recovery, dismantling, and remanufacturing processes are used to name each model; for example, in the SMM model, the seller handles the recovery and the manufacturer handles the last two. Theoretically, we did not mention three models (DMM, DMR, and SMR) because, in practice, (i) dismantling enterprises can and will dismantle scrapped products themselves; (ii) manufacturers who dismantle the scrapped products can and will handle the remanufacturing process as well; and (iii) remanufacturers frequently handle both dismantling and remanufacturing processes (to express generally, we still use "DR" to represent this situation). We present the following assumptions and notations for the five agricultural machinery remanufacturing models aforementioned.

**Table 2.** Models of current agricultural machinery recycling and remanufacturing channels.

| Model | Recovery | | Dismantling | | Remanufacturing | |
| --- | --- | --- | --- | --- | --- | --- |
| | Seller | Dismantling Enterprise | Manufacturer | Dismantling Enterprise | Manufacturer | Remanufacturer |
| SMM model | √ * | | √ | | √ | |
| SDM model | √ | | | √ | √ | |
| DDM model | | √ | | √ | √ | |
| SDR model | √ | | | √ | | √ |
| DDR model | | √ | | √ | | √ |

* The symbol "√" indicates the actor involved in each procedure.

### 3.2. Model Assumptions and Notations

Based on the preceding discussion, the following assumptions and notations are proposed to build the models.

#### 3.2.1. Assumptions

**Assumption 1.** *The system consists of a single manufacturer, seller, dismantling enterprise, and remanufacturer. All parties are rational and make decisions based on the principle of profit maximization.*

**Assumption 2.** *Agricultural machinery manufacturers and sellers are typically part of the same enterprise, so decisions are made collectively.*

**Assumption 3.** *New and remanufactured products are classified into different quality levels and are priced differently.*

**Assumption 4.** *The market demand for agricultural machinery exceeds the supply of new products; thus, there is market space for remanufactured products.*

**Assumption 5.** *Because scrapped products are not in perfect condition, not all recovered products can be remanufactured.*

**Assumption 6.** *In reality, each remanufacturer must have its own dismantling enterprise (department), but the dismantling enterprise is not always associated with one or more remanufacturers.*

### 3.2.2. Notations

For clarity, the notations for manufacturer, seller, dismantling enterprise, remanufacturer, and market are shown in Tables 3–7, respectively.

**Table 3.** Notation and definition for manufacturer.

| Notation | Definition |
|---|---|
| $c_{Mn}$ | unit production cost of new products for manufacturer |
| $c_{Mr}$ | unit production cost of remanufactured products for manufacturer, $c_{Mr} = c_{Mn} - \eta_M\theta_M$ or $c_{Mr} = c_{Mn} - \eta_M\theta_D$ |
| $c_{r,M}$ | unit dismantling cost for manufacturer |
| $\theta_M$ | the utilization rate of machine parts dismantled by manufacturer |
| $\eta_M$ | the influence factor of $\theta_M$ on $c_{Mr}$ |
| $p_{r,DM}$ | unit recovery price of available machine parts sold by dismantling enterprise to manufacturer |
| $\delta^x$ | unit transfer payment paid by manufacturer and seller to third-party enterprise, e.g., $x = \text{II}$ means the transfer payment in Model II |
| $C_{M1}$ | fixed production cost of new products for manufacturer |
| $C_{M2}$ | fixed dismantling cost for manufacturer |
| $C_{M3}$ | fixed production cost of remanufactured products for manufacturer |

**Table 4.** Notation and definition for seller.

| Notation | Definition |
|---|---|
| $p_n$ * | unit selling price of new products (decision variable) |
| $p_r$ | unit selling price of remanufactured products (decision variable) |
| $p_{r,S}$ | unit recovery price of scrapped products sold by consumer to seller (decision variable) |
| $c_{s,S}$ | unit storage cost of unused machine parts for seller |
| $C_{S1}$ | fixed selling cost for seller |
| $C_{S2}$ | fixed recovery cost of consumers' scrapped products for seller |

* We assumed that transportation costs are included in the prices.

**Table 5.** Notation and definition for dismantling enterprise.

| Notation | Definition |
|---|---|
| $p_{r,D}$ | unit recovery price of scrapped products sold by consumer to dismantling enterprise (decision variable) |
| $p_{r,SD}$ | unit recovery price of scrapped products sold by seller to dismantling enterprise |
| $c_{r,D}$ | unit dismantling cost for dismantling enterprise |
| $c_{s,D}$ | unit storage cost of unused machine parts for dismantling enterprise |
| $\theta_D$ | the utilization rate of machine parts dismantled by dismantling enterprise |
| $\eta_D$ | the influence factor of $\theta_D$ on $c_{Rr}$ |
| $C_{D1}$ | fixed dismantling cost for dismantling enterprise |
| $C_{D2}$ | fixed recovery cost of consumers' scrapped products for dismantling enterprise |

**Table 6.** Notation and definition for remanufacturer.

| Notation | Definition |
| --- | --- |
| $w_{Rr}$ | unit wholesale price of remanufactured products produced by remanufacturer |
| $c_{Rr}$ | unit production cost of remanufactured products for remanufacturer, $c_{Rr} = c_{Mn} - \eta_D \theta_D$ |
| $C_{R1}$ | fixed production cost of remanufactured products for remanufacturer |

**Table 7.** Notation and definition for market.

| Notation | Definition |
| --- | --- |
| $B$ | the possible largest market demand |
| $\alpha$ | sensitivity coefficient of new products' price to market capacity; for replaceable products, $\alpha < 1$; for irreplaceable products, $\alpha > 1$ |
| $q$ | market demand of new products, $q = -\alpha p_n + B$ |
| $\beta$ | influence coefficient of difference between prices of new and remanufactured products on market demand of remanufactured products; $\beta$ is nonnegative and positively correlated with quality of remanufactured products |
| $q_r$ | market demand of remanufactured products, $q_r = \beta(p_n - p_r)$ |
| $A$ | basic recovery quantity that does not depend on the recovery price |
| $\mu$ | influence coefficient of unit recovery price of scrapped products on recovery quantity, $\mu \geq 0$ |
| $Q_r$ | recovery quantity, $Q_r = A + \mu p_{r,S} + \varepsilon$ or $Q_r = A + \mu p_{r,D} + \varepsilon, \varepsilon \sim U[0, 1]$ |

## 4. Model Development

### 4.1. SMM Model (Model I)

Figure 1 depicts the CLSC system in the SMM model. In forward direction, manufacturer sells new and remanufactured products to consumers through sellers. In the opposite direction, the seller collects scrapped agricultural machines from consumers at a set price and returns them to the manufacturer. The manufacturer then dismantles them and reuses machine parts to produce remanufactured products. In this model, based on Assumption 2, all the decision variables, including unit selling price of new products $p_n$, unit selling price of remanufactured products $p_r$, and unit recovery price of scrapped products sold by consumer to seller $p_{r,S}$, are decided by manufacturer and seller. Accordingly, the total profit of manufacturer and seller is

$$\pi_M^I + \pi_S^I = (p_n - c_{Mn})q + (p_r - c_{Mr}) \cdot \min(q_r, \theta_M Q_r) - (p_{r,S} + c_{r,M})Q_r - c_{s,S}(\theta_M Q_r - q_r)^+ - C_{M1} - C_{M2} - C_{M3} - C_{S1} - C_{S2} \quad (1)$$

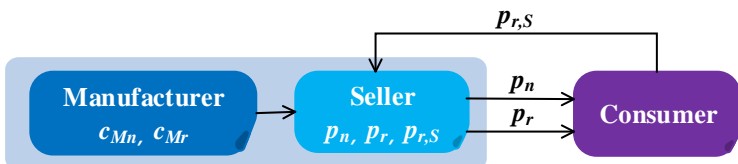

**Figure 1.** The framework of SMM model.

In this formula, the first item on the right is the profit from selling new products, the second item is the profit from selling remanufactured products, the third item is recovery and disassembly cost, the fourth item is storage cost, and the rest is fixed cost. Thus, when the supply of remanufactured products is less than demand, i.e., $\theta_M Q_r \leq q_r$, the total profit of manufacturer and seller is

$$\pi_M^I + \pi_S^I = (p_n - c_{Mn})q + (p_r - c_{Mr}) \cdot \theta_M Q_r - (p_{r,S} + c_{r,M})Q_r - C_{M1} - C_{M2} - C_{M3} - C_{S1} - C_{S2} \quad (2)$$

**Proposition 1.** *There is a unique optimal solution to Equation (2); in the SMM model, when the supply of remanufactured products is less than the demand, the optimal selling price of new products, the optimal selling price of remanufactured products, and the optimal recovery price of scrapped products are, respectively, as follows:*

$$p_n^{I*} = \frac{B + \alpha c_{Mn}}{2\alpha} \tag{3}$$

$$p_r^{I*} = \frac{2\beta p_n^{I*} - (A + \varepsilon)\theta_M + \mu\theta_M[(c_{Mn} - \eta_M\theta_M) \cdot \theta_M + c_{r,M}]}{2\beta + \mu\theta_M^2} \tag{4}$$

$$p_{r,S}^{I*} = \frac{\beta(p_n^{I*} - p_r^{I*})}{\mu\theta_M} - \frac{A + \varepsilon}{\mu} \tag{5}$$

Related proof is in Appendix A.

On the contrary, when the supply of remanufactured products is more than demand (i.e., $\theta_M Q_r > q_r$), the total profit of the manufacturer and seller is

$$\pi_M^I + \pi_S^I = (p_n - c_{Mn})q + (p_r - c_{Mr}) \cdot q_r - (p_{r,S} + c_{r,M})Q_r - c_{s,S}(\theta_M Q_r - q_r) - C_{M1} - C_{M2} - C_{M3} - C_{S1} - C_{S2} \tag{6}$$

**Proposition 2.** *There is a unique optimal solution to Formula (6); in the SMM model, when the supply of remanufactured products is greater than the demand, the optimal selling price of new products, the optimal selling price of remanufactured products, and the optimal recovery price of scrapped products are as follows:*

$$p_n^{I**} = \frac{2B + 2\alpha c_{Mn} + \beta(\eta_M\theta_M + c_{s,S} - c_{Mn})}{4\alpha - \beta} \tag{7}$$

$$p_r^{I**} = \frac{B + \alpha c_{Mn} - (2\alpha - \beta)(\eta_M\theta_M + c_{s,S} - c_{Mn})}{4\alpha - \beta} \tag{8}$$

$$p_{r,S}^{I**} = -\frac{c_{r,M} + c_{s,S}\theta_M}{2} - \frac{A + \varepsilon}{2\mu} \tag{9}$$

Related proof is in Appendix A.

*4.2. SDM Model (Model II)*

Figure 2 depicts the CLSC system in the SDM model. Manufacturers sell new and remanufactured products to consumers through sellers in the future. In reverse, the seller buys scrapped agricultural machines from consumers at a set price and sells them to a dismantling enterprise. The machines are then dismantled by the dismantling enterprise, which sells machine parts with resale value to the manufacturer. In this model, based on Assumption 2, all the decision variables, including unit selling price of new products $p_n$, unit selling price of remanufactured products $p_r$, and unit recovery price of scrapped products sold by consumer to seller $p_{r,S}$, are decided by manufacturer and seller. Accordingly, the total profit of manufacturer and seller is

$$\pi_M^{II} + \pi_S^{II} = (p_n - c_{Mn})q + (p_r - c_{Mr}) \cdot \min(q_r, \theta_D Q_r) - (p_{r,S} + \theta_D p_{r,DM} - p_{r,SD})Q_r - c_{s,S}(\theta_D Q_r - q_r)^+ \\ - C_{M1} - C_{M3} - C_{S1} - C_{S2} \tag{10}$$

Affected by the decision of manufacturer and seller, the profit of dismantling enterprise is

$$\pi_D^{II} = (\theta_D p_{r,DM} - p_{r,SD} - c_{r,D})Q_r - C_{D1} \tag{11}$$

For manufacturer and seller, in Equation (10), the first item on the right is the profit from selling new products, the second item is the profit from selling remanufactured products, the third item is recovery and disassembly cost, the fourth item is storage cost,

and the rest is fixed cost. For dismantling enterprise, in Equation (11), the first item on the right is the profit from dismantling scrapped products and the rest is fixed cost. In this case, the transfer payment $\delta^{\mathrm{II}} = \theta_D p_{r,DM} - p_{r,SD}$.

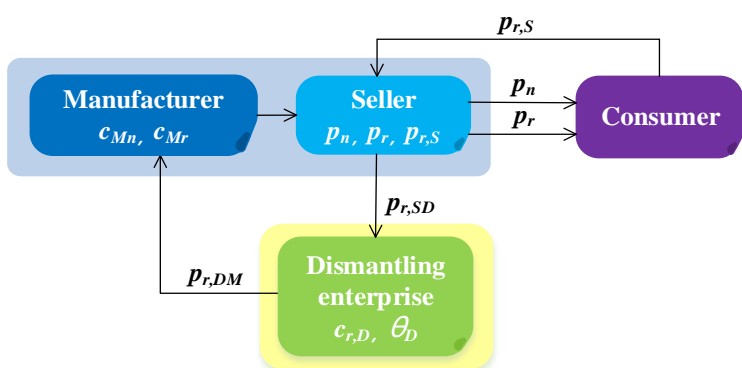

**Figure 2.** The framework of SDM model.

Similar to the derivation process in Section 4.1, we can obtain the derivation results of the SDM model below. The proof process in Appendix A can be used as the reference. When the supply of remanufactured products is less than demand (i.e., $\theta_D Q_r \le q_r$), the total profit of manufacturer and seller is

$$\pi_M^{\mathrm{II}} + \pi_S^{\mathrm{II}} = (p_n - c_{Mn})q + (p_r - c_{Mr}) \cdot \theta_D Q_r - (p_{r,S} + \theta_D p_{r,DM} - p_{r,SD})Q_r - C_{M1} - C_{M3} - C_{S1} - C_{S2} \tag{12}$$

The profit of dismantling enterprise is

$$\pi_D^{\mathrm{II}} = (\theta_D p_{r,DM} - p_{r,SD} - c_{r,D})Q_r - C_{D1} \tag{13}$$

**Proposition 3.** *There is a unique optimal solution to Equation (12); in the SDM model, when the supply of remanufactured products is less than the demand, the optimal selling price of new products, the optimal selling price of remanufactured products, and the optimal recovery price of scrapped products are, respectively, as follows:*

$$p_n^{\mathrm{II}*} = \frac{B + \alpha c_{Mn}}{2\alpha} \tag{14}$$

$$p_r^{\mathrm{II}*} = \frac{\mu \theta_D [(c_{Mn} - \eta_M \theta_D) \cdot \theta_D + (\theta_D p_{r,DM} - p_{r,SD})] + 2\beta p_n^{\mathrm{II}*} - (A + \varepsilon)\theta_D}{2\beta + \mu \theta_D^2} \tag{15}$$

$$p_{r,S}^{\mathrm{II}*} = \frac{\beta(p_n^{\mathrm{II}*} - p_r^{\mathrm{II}*})}{\mu \theta_D} - \frac{A + \varepsilon}{\mu} \tag{16}$$

Furthermore, we can obtain profits and the optimal solution when the supply of remanufactured products exceeds the demand (i.e., $\theta_D Q_r > q_r$):

$$\pi_M^{\mathrm{II}} + \pi_S^{\mathrm{II}} = (p_n - c_{Mn})q + (p_r - c_{Mr}) \cdot q_r - (p_{r,S} + \theta_D p_{r,DM} - p_{r,SD})Q_r - c_{s,S}(\theta_D Q_r - q_r) - C_{M1} - C_{M3} \\ - C_{S1} - C_{S2} \tag{17}$$

$$\pi_D^{\mathrm{II}} = (\theta_D p_{r,DM} - p_{r,SD} - c_{r,D})Q_r - C_{D1} \tag{18}$$

**Proposition 4.** *There is a unique optimal solution to Equation (17); in the SDM model, when the supply of remanufactured products is greater than the demand, the optimal selling price of new*

*products, the optimal selling price of remanufactured products, and the optimal recovery price of scrapped products are as follows*:

$$p_n^{\text{II}**} = \frac{\beta(\eta_M\theta_D + c_{s,S} - c_{Mn})}{4\alpha - \beta} + \frac{2B + 2\alpha c_{Mn}}{4\alpha - \beta} \tag{19}$$

$$p_r^{\text{II}**} = -\frac{(2\alpha - \beta)(\eta_M\theta_D + c_{s,S} - c_{Mn})}{4\alpha - \beta} + \frac{B + \alpha c_{Mn}}{4\alpha - \beta} \tag{20}$$

$$p_{r,S}^{\text{II}**} = -\frac{\theta_D p_{r,DM} - p_{r,SD} + c_{s,S}\theta_D}{2} - \frac{A + \varepsilon}{2\mu} \tag{21}$$

### 4.3. DDM Model (Model III)

Figure 3 depicts the DDM model's CLSC system. In forward direction, the manufacturers sell new and remanufactured products to consumers through sellers in the future. In the opposite direction, a dismantling enterprise collects scrapped agricultural machines from consumers for a fee. Following dismantling, the dismantling enterprise sells machine parts with reuse value to the manufacturer. In this model, according to Assumption 2, the decision variables decided by manufacturer and seller are unit selling price of new products $p_n$ and unit selling price of remanufactured products $p_r$. Accordingly, the total profit of manufacturer and seller is

$$\pi_M^{\text{III}} + \pi_S^{\text{III}} = (p_n - c_{Mn})q + (p_r - c_{Mr}) \cdot \min(q_r, \theta_D Q_r) - p_{r,DM} \cdot \min(q_r, \theta_D Q_r) - C_{M1} - C_{M3} - C_{S1} \tag{22}$$

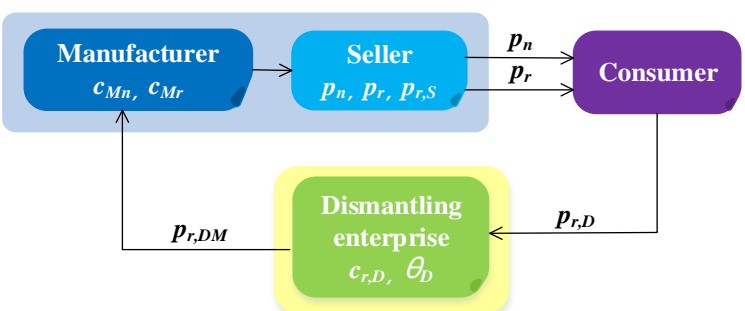

**Figure 3.** The framework of DDM model.

Based on Assumption 6, the decision variable decided by dismantling enterprise is the unit recovery price of scrapped products sold by consumers to the dismantling enterprise $p_{r,D}$. Therefore, the profit of dismantling enterprise is

$$\pi_D^{\text{III}} = p_{r,DM} \cdot \min(q_r, \theta_D Q_r) - (p_{r,D} + c_{r,D})Q_r - c_{s,D}(\theta_D Q_r - q_r)^+ - C_{D1} - C_{D2} \tag{23}$$

For manufacturer and seller, in Equation (22), the first item on the right is the profit from selling new products, the second item is the profit from selling remanufactured products, the third item is recovery and disassembly cost, and the rest is fixed cost. For dismantling enterprise, in Equation (23), the first item on the right is the profit from recovering and dismantling scrapped products, the second item is recovery and disassembly cost, the third item is storage cost, and the rest is fixed cost. In this case, the transfer payment $\delta^{\text{III}} = \theta_D p_{r,DM}$ when $\theta_D Q_r \leq q_r$ and $\delta^{\text{III}} = p_{r,DM}$ when $\theta_D Q_r > q_r$.

Similar to the derivation process in Section 4.1, we can obtain the derivation results of the DDM model below. The proof process in Appendix A can be used as the reference. When the supply of remanufactured products is less than demand (i.e., $\theta_D Q_r \leq q_r$), the total profit of the manufacturer and seller is

$$\pi_M^{\text{III}} + \pi_S^{\text{III}} = (p_n - c_{Mn})q + (p_r - c_{Mr}) \cdot \theta_D Q_r - p_{r,DM} \cdot \theta_D Q_r - C_{M1} - C_{M3} - C_{S1} \tag{24}$$

The profit of dismantling enterprise is

$$\pi_D^{\text{III}} = p_{r,DM} \cdot \theta_D Q_r - (p_{r,D} + c_{r,D})Q_r - C_{D1} - C_{D2} \tag{25}$$

**Proposition 5.** *There is a unique optimal solution to Equations (24) and (25); in the DDM model, when the supply of remanufactured products is less than the demand, the optimal selling price of new products, the optimal selling price of remanufactured products, and the optimal recovery price of scrapped products are, respectively, as follows:*

$$p_n^{\text{III}*} = \frac{B + \alpha c_{Mn}}{2\alpha} \tag{26}$$

$$p_r^{\text{III}*} = p_n^{\text{III}*} - \frac{\theta_D \left(A + \mu p_{r,D}^{\text{III}*} + \varepsilon\right)}{\beta} \tag{27}$$

$$p_{r,D}^{\text{III}*} = \frac{\theta_D p_{r,DM} - c_{r,D}}{2} - \frac{A + \varepsilon}{2\mu} \tag{28}$$

Furthermore, we can obtain profits and the best solution when the supply of remanufactured products exceeds the demand (i.e., $\theta_D Q_r > q_r$):

$$\pi_M^{\text{III}} + \pi_S^{\text{III}} = (p_n - c_{Mn})q + (p_r - c_{Mr}) \cdot q_r - p_{r,DM} \cdot q_r - C_{M1} - C_{M3} - C_{S1} \tag{29}$$

$$\pi_D^{\text{III}} = p_{r,DM} \cdot q_r - (p_{r,D} + c_{r,D})Q_r - c_{s,D}(\theta_D Q_r - q_r) - C_{D1} - C_{D2} \tag{30}$$

**Proposition 6.** *There is a unique optimal solution to Equations (29) and (30); in the DDM model, when the supply of remanufactured products is greater than the demand, the optimal selling price of new products, the optimal selling price of remanufactured products, and the optimal recovery price of scrapped products are as follows:*

$$p_n^{\text{III}**} = \frac{\beta(\eta_M \theta_D - c_{Mn} - p_{r,DM})}{4\alpha - \beta} + \frac{2B + 2\alpha c_{Mn}}{4\alpha - \beta} \tag{31}$$

$$p_r^{\text{III}**} = -\frac{(2\alpha - \beta)(\eta_M \theta_D - c_{Mn} - p_{r,DM})}{4\alpha - \beta} + \frac{B + \alpha c_{Mn}}{4\alpha - \beta} \tag{32}$$

$$p_{r,D}^{\text{III}**} = -\frac{c_{r,D} + c_{s,D}\theta_D}{2} - \frac{A + \varepsilon}{2\mu} \tag{33}$$

*4.4. SDR Model (Model IV)*

Figure 4 depicts the CLSC system in the SDR model. In forward, the manufacturer sells new products and the remanufacturer sells remanufactured products to consumers via the seller. In reverse, the seller buys scrapped agricultural machines from consumers at a set price and sells them to a dismantling enterprise. After dismantling, the dismantling enterprise sells reusable machine parts to a remanufacturer. The remanufacturer creates remanufactured products and sells them to consumers via the seller. In this model, based on Assumption 2, all the decision variables, including unit selling price of new products $p_n$, unit selling price of remanufactured products $p_r$, and unit recovery price of scrapped products $p_{r,S}$, are decided by manufacturer and seller. Accordingly, the total profit of manufacturer and seller is

$$\pi_M^{\text{IV}} + \pi_S^{\text{IV}} = (p_n - c_{Mn})q + p_r \cdot \min(q_r, \theta_D Q_r) + (p_{r,SD} - \theta_D w_{Rr} - p_{r,S})Q_r - c_{s,S}(\theta_D Q_r - q_r)^+ - C_{M1} - C_{S1} - C_{S2} \tag{34}$$

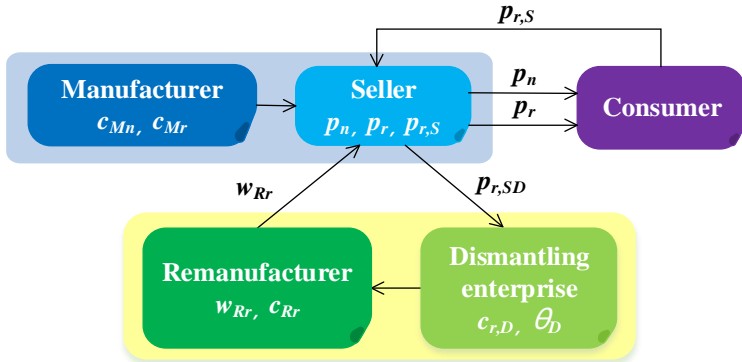

**Figure 4.** The framework of SDR model.

In this case, according to model assumptions, dismantling enterprise is an associated enterprise of remanufacturer, and, affected by the decision of manufacturer and seller, the total profit of dismantling enterprise and remanufacturer is

$$\pi_D^{\text{IV}} + \pi_R^{\text{IV}} = (w_{Rr} - c_{Rr})\theta_D Q_r - (p_{r,SD} + c_{r,D})Q_r - C_{D1} - C_{R1} \tag{35}$$

For manufacturer and seller, in Equation (34), the first item on the right is the profit from selling new products, the second item is the profit from selling remanufactured products, the third item is recovery and remanufacturing cost, the fourth item is storage cost, and the rest is fixed cost. For dismantling enterprise and remanufacturer, in Equation (35), the first item on the right is the profit from remanufacturing, the second item is disassembly and remanufacturing cost, and the rest is fixed cost. In this case, the transfer payment $\delta^{\text{IV}} = \theta_D w_{Rr} - p_{r,SD}$.

Similar to the derivation process in Section 4.1, we have the derivation results of the SDR model below. The proof process in Appendix A can be used as the reference. When the supply of remanufactured products is less than demand (i.e., $\theta_D Q_r \le q_r$), the total profit of manufacturer and seller is

$$\pi_M^{\text{IV}} + \pi_S^{\text{IV}} = (p_n - c_{Mn})q + p_r \cdot \theta_D Q_r + (p_{r,SD} - \theta_D w_{Rr} - p_{r,S})Q_r - C_{M1} - C_{S1} - C_{S2} \tag{36}$$

The total profit of dismantling enterprise and remanufacturer is

$$\pi_D^{\text{IV}} + \pi_R^{\text{IV}} = (w_{Rr} - c_{Rr})\theta_D Q_r - (p_{r,SD} + c_{r,D})Q_r - C_{D1} - C_{R1} \tag{37}$$

**Proposition 7.** *There is a unique optimal solution to Equation (36); in the SDR model, when the supply of remanufactured products is less than the demand, the optimal selling price of new products, the optimal selling price of remanufactured products, and the optimal recovery price of scrapped products are, respectively, as follows:*

$$p_n^{\text{IV}*} = \frac{B + \alpha c_{Mn}}{2\alpha} \tag{38}$$

$$p_r^{\text{IV}*} = \frac{\mu \theta_D(w_{Rr} \cdot \theta_D - p_{r,SD}) + 2\beta p_n^{\text{IV}*} - (A + \varepsilon)\theta_D}{2\beta + \mu \theta_D^2} \tag{39}$$

$$p_{r,S}^{\text{IV}*} = \frac{\beta(p_n^{\text{IV}*} - p_r^{\text{IV}*})}{\mu \theta_D} - \frac{A + \varepsilon}{\mu} \tag{40}$$

Furthermore, we can obtain profits and the optimal solution when the supply of remanufactured products exceeds the demand (i.e., $\theta_D Q_r > q_r$):

$$\pi_M^{\text{IV}} + \pi_S^{\text{IV}} = (p_n - c_{Mn})q + p_r \cdot q_r + (p_{r,SD} - \theta_D w_{Rr} - p_{r,S})Q_r - c_{s,S}(\theta_D Q_r - q_r) - C_{M1} - C_{S1} - C_{S2} \tag{41}$$

$$\pi_D^{\text{IV}} + \pi_R^{\text{IV}} = (w_{Rr} - c_{Rr})\theta_D Q_r - (p_{r,SD} + c_{r,D})Q_r - C_{D1} - C_{R1} \tag{42}$$

**Proposition 8.** *There is a unique optimal solution to Equation (41); in the SDR model, when the supply of remanufactured products is greater than the demand, the optimal selling price of new products, the optimal selling price of remanufactured products, and the optimal recovery price of scrapped products are as follows:*

$$p_n^{\text{IV}**} = \frac{c_{s,S} \cdot \beta}{4\alpha - \beta} + \frac{2B + 2\alpha c_{Mn}}{4\alpha - \beta} \tag{43}$$

$$p_r^{\text{IV}**} = -\frac{c_{s,S}(2\alpha - \beta)}{4\alpha - \beta} + \frac{B + \alpha c_{Mn}}{4\alpha - \beta} \tag{44}$$

$$p_{r,S}^{\text{IV}**} = -\frac{\theta_D w_{Rr} - p_{r,SD} + c_{s,S}\theta_D}{2} - \frac{A + \varepsilon}{2\mu} \tag{45}$$

*4.5. DDR Model (Model V)*

Figure 5 depicts the DDR model's CLSC system. In forward direction, the manufacturer sells new products and the remanufacturer sells remanufactured products to consumers via the seller. In the opposite direction, a dismantling enterprise collects scrapped agricultural machines from consumers for a fee. After dismantling, the dismantling enterprise sells reusable machine parts to a remanufacturer. The remanufacturer creates remanufactured products and sells them to consumers via the seller. Based on Assumption 2, the decision variables of the manufacturer and seller in this model are the unit selling price of new products $p_n$ and unit selling price of remanufactured products $p_r$. Accordingly, the total profit of manufacturer and seller is

$$\pi_M^{\text{V}} + \pi_S^{\text{V}} = (p_n - c_{Mn})q + (p_r - w_{Rr}) \cdot \min(q_r, \theta_D Q_r) - C_{M1} - C_{S1} \tag{46}$$

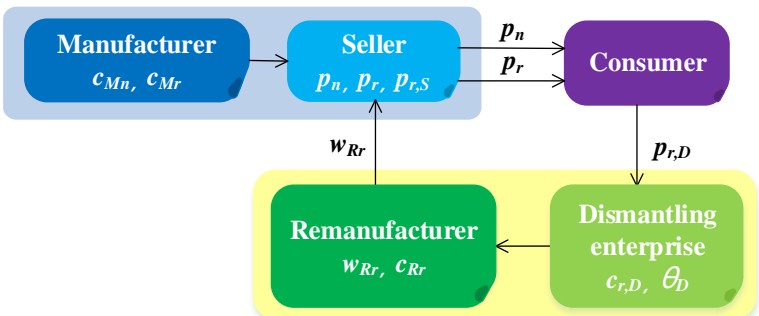

**Figure 5.** The framework of DDR model.

In this case, according to Assumption 6, dismantling enterprise is an associated enterprise of remanufacturer, and the decision variable of them is the unit recovery price of scrapped products sold by the consumer to the dismantling enterprise $p_{r,D}$. Therefore, the total profit of dismantling enterprise and remanufacturer is

$$\pi_D^{\text{V}} + \pi_R^{\text{V}} = (w_{Rr} - c_{Rr}) \cdot \min(q_r, \theta_D Q_r) - (p_{r,D} + c_{r,D})Q_r - c_{s,D}(\theta_D Q_r - q_r)^+ - C_{D1} - C_{D2} - C_{R1} \tag{47}$$

For manufacturer and seller, in Equation (46), the first item on the right is the profit from selling new products, the second item is the profit from selling remanufactured products, and the rest is fixed cost. For dismantling enterprise and remanufacturer, in Equation (47), the first item on the right is the profit from remanufacturing, the second item is recovery and disassembly cost, the third item is storage cost, and the rest is fixed cost. In this case, the transfer payment $\delta^{\text{V}} = \theta_D w_{Rr}$ when $\theta_D Q_r \leq q_r$ and $\delta^{\text{V}} = w_{Rr}$ when $\theta_D Q_r > q_r$.

Similar to the derivation process in Section 4.1, we have the derivation results of the DDR model below. The proof process in Appendix A can be used as the reference. When the supply of remanufactured products is less than demand (i.e., $\theta_D Q_r \leq q_r$), the total profit of manufacturer and seller is

$$\pi_M^V + \pi_S^V = (p_n - c_{Mn})q + (p_r - w_{Rr}) \cdot \theta_D Q_r - C_{M1} - C_{S1} \tag{48}$$

The total profit of dismantling enterprise and remanufacturer is

$$\pi_D^V + \pi_R^V = (w_{Rr} - c_{Rr}) \cdot \theta_D Q_r - (p_{r,D} + c_{r,D})Q_r - C_{D1} - C_{D2} - C_{R1} \tag{49}$$

**Proposition 9.** *There is a unique optimal solution to Equations (48) and (49); in the DDR model, when the supply of remanufactured products is less than the demand, the optimal selling price of new products, the optimal selling price of remanufactured products, and the optimal recovery price of scrapped products are, respectively, as follows:*

$$p_n^{V*} = \frac{B + \alpha c_{Mn}}{2\alpha} \tag{50}$$

$$p_r^{V*} = p_n^{V*} - \frac{\theta_D \left( A + \mu p_{r,D}^{V*} + \varepsilon \right)}{\beta} \tag{51}$$

$$p_{r,D}^{V*} = \frac{(w_{Rr} - c_{Mn} + \eta_D \theta_D) \cdot \theta_D - c_{r,D}}{2} - \frac{A + \varepsilon}{2\mu} \tag{52}$$

Furthermore, we can obtain profits and the optimal solution when the supply of remanufactured products exceeds the demand (i.e., $\theta_D Q_r > q_r$):

$$\pi_M^V + \pi_S^V = (p_n - c_{Mn})q + (p_r - w_{Rr}) \cdot q_r - C_{M1} - C_{S1} \tag{53}$$

$$\pi_D^V + \pi_R^V = (w_{Rr} - c_{Rr}) \cdot q_r - (p_{r,D} + c_{r,D})Q_r - c_{s,D}(\theta_D Q_r - q_r) - C_{D1} - C_{D2} - C_{R1} \tag{54}$$

**Proposition 10.** *There is a unique optimal solution to Equations (53) and (54); in the DDR model, when the supply of remanufactured products is greater than the demand, the optimal selling price of new products, the optimal selling price of remanufactured products, and the optimal recovery price of scrapped products are as follows:*

$$p_n^{V**} = \frac{-w_{Rr} \cdot \beta}{4\alpha - \beta} + \frac{2B + 2\alpha c_{Mn}}{4\alpha - \beta} \tag{55}$$

$$p_r^{V**} = \frac{w_{Rr}(2\alpha - \beta)}{4\alpha - \beta} + \frac{B + \alpha c_{Mn}}{4\alpha - \beta} \tag{56}$$

$$p_{r,D}^{V**} = -\frac{c_{r,D} + c_{s,D}\theta_D}{2} - \frac{A + \varepsilon}{2\mu} \tag{57}$$

### 4.6. Model Comparison

Based on the supply and demand of remanufactured products, the equilibrium results of each model discussed above are summarized in Tables 8 and 9.

As shown in Table 8, when the supply of remanufactured products is less than the demand, (i) in SMM model, SDM model, and SDR model, the optimal selling prices of new and remanufactured products meet the condition $p_n - p_r \geq \frac{(A+\varepsilon)\cdot\theta}{\beta}$, $\theta \in \{\theta_M, \theta_D\}$ based on the assumption $p_{r,S} \geq 0$. Under such condition, consumers are willing to sell the scrapped agricultural machines to seller and buy remanufactured products; (ii) the optimal selling prices of new products $p_n$ are the same in all the five models, which are influenced

by market demand, sensitivity coefficient of new products' price to market capacity, and new products' production cost.

**Table 8.** Equilibrium results of each model when the supply of remanufactured products is less than demand.

| Recycling Channel Model | $p_n$ | $p_r$ | $p_{r,S}/p_{r,D}$ |
|---|---|---|---|
| SMM model | $p_n^{I*} = \frac{B+\alpha c_{Mn}}{2\alpha}$ | $p_r^{I*} = \frac{\mu\theta_M\left[(c_{Mn}-\eta_M\theta_M)\cdot\theta_M+c_{r,M}\right]+2\beta p_n^{I*}-(A+\varepsilon)\theta_M}{2\beta+\mu\theta_M^2}$ | $p_{r,S}^{I*} = \frac{\beta(p_n^{I*}-p_r^{I*})}{\mu\theta_M} - \frac{A+\varepsilon}{\mu}$ |
| SDM model | $p_n^{II*} = \frac{B+\alpha c_{Mn}}{2\alpha}$ | $p_r^{II*} = \frac{\mu\theta_D\left[(c_{Mn}-\eta_M\theta_D)\cdot\theta_D+(\theta_D p_{r,DM}-p_{r,SD})\right]+2\beta p_n^{II*}-(A+\varepsilon)\theta_D}{2\beta+\mu\theta_D^2}$ | $p_{r,S}^{II*} = \frac{\beta(p_n^{II*}-p_r^{II*})}{\mu\theta_D} - \frac{A+\varepsilon}{\mu}$ |
| DDM model | $p_n^{III*} = \frac{B+\alpha c_{Mn}}{2\alpha}$ | $p_r^{III*} = p_n^{III*} - \frac{\theta_D\left(A+\mu p_{r,D}^{III*}+\varepsilon\right)}{\beta}$ | $p_{r,D}^{III*} = \frac{\theta_D p_{r,DM}-c_{r,D}}{2} - \frac{A+\varepsilon}{2\mu}$ |
| SDR model | $p_n^{IV*} = \frac{B+\alpha c_{Mn}}{2\alpha}$ | $p_r^{IV*} = \frac{\mu\theta_D(w_{Rr}\cdot\theta_D-p_{r,SD})+2\beta p_n^{IV*}-(A+\varepsilon)\theta_D}{2\beta+\mu\theta_D^2}$ | $p_{r,S}^{IV*} = \frac{\beta(p_n^{IV*}-p_r^{IV*})}{\mu\theta_D} - \frac{A+\varepsilon}{\mu}$ |
| DDR model | $p_n^{V*} = \frac{B+\alpha c_{Mn}}{2\alpha}$ | $p_r^{V*} = p_n^{V*} - \frac{\theta_D\left(A+\mu p_{r,D}^{V*}+\varepsilon\right)}{\beta}$ | $p_{r,D}^{V*} = \frac{(w_{Rr}-c_{Mn}+\eta_D\theta_D)\cdot\theta_D-c_{r,D}}{2} - \frac{A+\varepsilon}{2\mu}$ |

**Table 9.** Equilibrium results of each model when the supply of remanufactured products is more than demand.

| Recycling Channel Model | $p_n$ | $p_r$ | $p_{r,S}/p_{r,D}$ |
|---|---|---|---|
| SMM model | $p_n^{I**} = \frac{\beta(\eta_M\theta_M+c_{s,S}-c_{Mn})}{4\alpha-\beta} + \frac{2B+2\alpha c_{Mn}}{4\alpha-\beta}$ | $p_r^{I**} = -\frac{(2\alpha-\beta)(\eta_M\theta_M+c_{s,S}-c_{Mn})}{4\alpha-\beta} + \frac{B+\alpha c_{Mn}}{4\alpha-\beta}$ | $p_{r,S}^{I**} = -\frac{c_{r,M}+c_{s,S}\theta_M}{2} - \frac{A+\varepsilon}{2\mu}$ |
| SDM model | $p_n^{II**} = \frac{\beta(\eta_M\theta_D+c_{s,S}-c_{Mn})}{4\alpha-\beta} + \frac{2B+2\alpha c_{Mn}}{4\alpha-\beta}$ | $p_r^{II**} = -\frac{(2\alpha-\beta)(\eta_M\theta_D+c_{s,S}-c_{Mn})}{4\alpha-\beta} + \frac{B+\alpha c_{Mn}}{4\alpha-\beta}$ | $p_{r,S}^{II**} = -\frac{\theta_D p_{r,DM}-p_{r,SD}+c_{s,S}\theta_D}{2} - \frac{A+\varepsilon}{2\mu}$ |
| DDM model | $p_n^{III**} = \frac{\beta(\eta_M\theta_D-c_{Mn}-p_{r,DM})}{4\alpha-\beta} + \frac{2B+2\alpha c_{Mn}}{4\alpha-\beta}$ | $p_r^{III**} = -\frac{(2\alpha-\beta)(\eta_M\theta_D+c_{s,S}-c_{Mn})}{4\alpha-\beta} + \frac{B+\alpha c_{Mn}}{4\alpha-\beta}$ | $p_{r,D}^{III**} = -\frac{c_{r,D}+c_{s,D}\theta_D}{2} - \frac{A+\varepsilon}{2\mu}$ |
| SDR model | $p_n^{IV**} = \frac{c_{s,S}\cdot\beta}{4\alpha-\beta} + \frac{2B+2\alpha c_{Mn}}{4\alpha-\beta}$ | $p_r^{IV**} = -\frac{c_{s,S}(2\alpha-\beta)}{4\alpha-\beta} + \frac{B+\alpha c_{Mn}}{4\alpha-\beta}$ | $p_{r,S}^{IV**} = -\frac{\theta_D w_{Rr}-p_{r,SD}+c_{s,S}\theta_D}{2} - \frac{A+\varepsilon}{2\mu}$ |
| DDR model | $p_n^{V**} = \frac{-w_{Rr}\cdot\beta}{4\alpha-\beta} + \frac{2B+2\alpha c_{Mn}}{4\alpha-\beta}$ | $p_r^{V**} = \frac{w_{Rr}(2\alpha-\beta)}{4\alpha-\beta} + \frac{B+\alpha c_{Mn}}{4\alpha-\beta}$ | $p_{r,D}^{V**} = -\frac{c_{r,D}+c_{s,D}\theta_D}{2} - \frac{A+\varepsilon}{2\mu}$ |

As shown in Table 9, when the supply of remanufactured products is more than demand, (i) the optimal selling prices of new and remanufactured products meet the condition $p_n - p_r = \frac{B+\alpha c_{Mn}}{4\alpha} + \frac{2\alpha\cdot\Delta}{4\alpha-\beta}$, $\Delta = \left\{\Delta^I, \Delta^{II}, \Delta^{III}, \Delta^{IV}, \Delta^V\right\}$. Among them, $\Delta^I = \eta_M\theta_M + c_{s,S} - c_{Mn}$, $\Delta^{II} = \eta_M\theta_D + c_{s,S} - c_{Mn}$, $\Delta^{III} = \eta_M\theta_D - c_{Mn} - p_{r,DM}$, $\Delta^{IV} = c_{s,S}$, $\Delta^V = -w_{Rr}$; (ii) in both the DDM and DDR models, the optimal recovery price of scrapped products $p_{r,D}$ is the same. However, since the transfer payments are positive, the optimal recovery prices of scrapped products $p_{r,S}/p_{r,D}$ are negative in all five models, which may lead to disruptions in recycling and remanufacturing.

## 5. Numerical Analysis

This study aims to explore the influence of various factors on decision variables and expected profits more intuitively. Based on the preceding discussion and data from the agricultural machinery recycling and remanufacturing industries in Zhejiang Province, we assign reasonable values to the relevant variable parameters of the above models to further analyze the applicable conditions of each model. As a major economic province with limited resources, Zhejiang Province is a key area for the remanufacturing industry [54]. Statistics show that the number of privately owned agricultural machines in Zhejiang Province reached 66,318 by the end of 2020. With a scrapping rate of 3% (6–10% in developed countries), Zhejiang Province will need to scrap approximately 2000 agricultural machines each year. As a result, we conducted a one-month investigation into Zhejiang Province's agricultural machinery recycling and remanufacturing industry.

According to our investigation, we assume the values of model variables and parameters as shown in Table 10. Thus, the optimal values of the decision variables and the optimal profits of supply chain members in different models are obtained through preceding discussion. Furthermore, the transfer payments made by the manufacturer and seller to the third-party enterprise are the primary benefit of the third-party enterprise and the primary deciding factor in the selection of each model. Therefore, the following discussion is based on the transfer payment to discuss the applicable conditions of different models, with the SMM model (manufacturer and seller undertake the entire recycling and

remanufacturing process themselves) serving as a comparison point. For clarity, we will first discuss the situation in which the supply of remanufactured products is less than the demand, and then we will discuss the opposite situation.

**Table 10.** Assumed values for model variables and parameters.

| Notation | Value [1] | Notation | Value | Notation | Value |
|---|---|---|---|---|---|
| $A$ | 20 | $C_{D2}$ | CNY 1000 | $\eta_D$ | 35 |
| $B$ | 500 | $C_{R1}$ | CNY 1500 | $\theta_M$ | 0.5 |
| $C_{M1}$ | CNY 10,000 [2] | $c_{Mn}$ | CNY 30 | $\theta_D$ | 0.8 |
| $C_{M2}$ | CNY 1000 | $c_{r,M}$ | CNY 1 | $\varepsilon$ | 0.5 |
| $C_{M3}$ | CNY 2000 | $c_{r,D}$ | CNY 0.1 | $\alpha$ | 1 |
| $C_{S1}$ | CNY 5000 | $c_{s,S}$ | CNY 0.05 | $\beta$ | 3 |
| $C_{S2}$ | CNY 1000 | $c_{s,D}$ | CNY 0.04 | $\mu$ | 5 |
| $C_{D1}$ | CNY 2000 | $\eta_M$ | 30 | | |

[1] Due to the large variety of agricultural machinery, this paper roughly selected the average value of common agricultural machinery products within a certain time range according to the investigation, which is used to reflect the actual relationship between the model variables and parameters, but it is not the exact value of the industry, which is hereby explained. [2] Yuan is the generic unit of Chinese currency RMB.

### 5.1. Applicable Conditions of the SMM, SDM, and DDM Models

In this section, we first focus on a CLSC without remanufacturer and discuss the applicable conditions of the SMM, SDM, and DDM models based on the effect of transfer payments paid by manufacturer and seller on the optimal profits of members when the supply of remanufactured products is less than demand (Figure 6). The transfer payments in the SDM and DDM models meet $\delta^{II} = \theta_D p_{r,DM} - p_{r,SD}$ and $\delta^{III} = \theta_D p_{r,DM}$, respectively.

Figure 6 shows that the optimal profit of all entities in each model varies with a change in transfer payments. As shown in Figure 6, when there is no remanufacturer, an increase in transfer payments is not always beneficial to the dismantling enterprise or other entities. However, a reasonable transfer payment can create a win–win situation for all supply chain members. Specifically, when the value of the transfer payment $\delta^{II}$ is between 6.08 and 54.68, the total profit of the manufacturer and seller in the SDM model is greater than that in the SMM model, and the profit of the dismantling enterprise is nonnegative, ensuring that all parties are willing to participate. Therefore, if the value of the transfer payment $\delta$ is between 6.08 and 54.68, compared with the SMM model, supply chain members would be better off using the SDM model. Similarly, when the transfer payment $\delta^{III}$ value is between 44.99 and 104.93, the total profit of the manufacturer and seller is higher in the DDM model than in the SMM model, and the profit of the dismantling enterprise is nonnegative. Therefore, if the transfer payment $\delta$ value is between 44.99 and 104.93, compared with the SMM model, the DDM model is a better option for supply chain members. Note that the results of the preceding analysis also indicate the applicable transfer payment range in the specific model. As a representative example, the MATLAB code and dataset for applicable conditions of the SMM and SDM models are presented in the Supplementary Materials file.

### 5.2. Applicable Conditions of the SMM, SDR, and DDR Models

This section focuses on a CLSC with a remanufacturer and discusses the applicable conditions of the SMM, SDR, and DDR models based on the effect of transfer payments paid by the manufacturer and seller on the optimal profits of members when the supply of remanufactured products is less than demand (Figure 7). The transfer payment in the SDR model meets $\delta^{IV} = \theta_D w_{Rr} - p_{r,SD}$, and the transfer payment in the DDR model meets $\delta^{V} = \theta_D w_{Rr}$.

Figure 7 shows that the optimal profit of all entities in each model varies with change in transfer payments. As in the previous section, a reasonable transfer payment can result in a win–win situation for all supply chain members when there are remanufacturers in the supply chain. Specifically, when the value of the transfer payment $\delta^{IV}$ is between 12.23 and 71.99, the total profit of the manufacturer and seller in the SDR model is greater than that

in the SMM model, and the profit of the dismantling enterprise is nonnegative, ensuring that all parties are willing to participate. Therefore, if the value of the transfer payment $\delta$ is between 12.23 and 71.99, compared with the SMM model, supply chain members would be better off using the SDR model. Similarly, when the transfer payment $\delta^V$ value is between 57.60 and 115.18, the total profit of the manufacturer and seller is higher in the DDR model than in the SMM model, and the profit of the dismantling enterprise is nonnegative. As a result, if the transfer payment $\delta$ value is between 57.60 and 115.18, compared with the SMM model, the DDR model is a better option for supply chain members. Note that the results of the preceding analysis also indicate the applicable transfer payment range in the specific model.

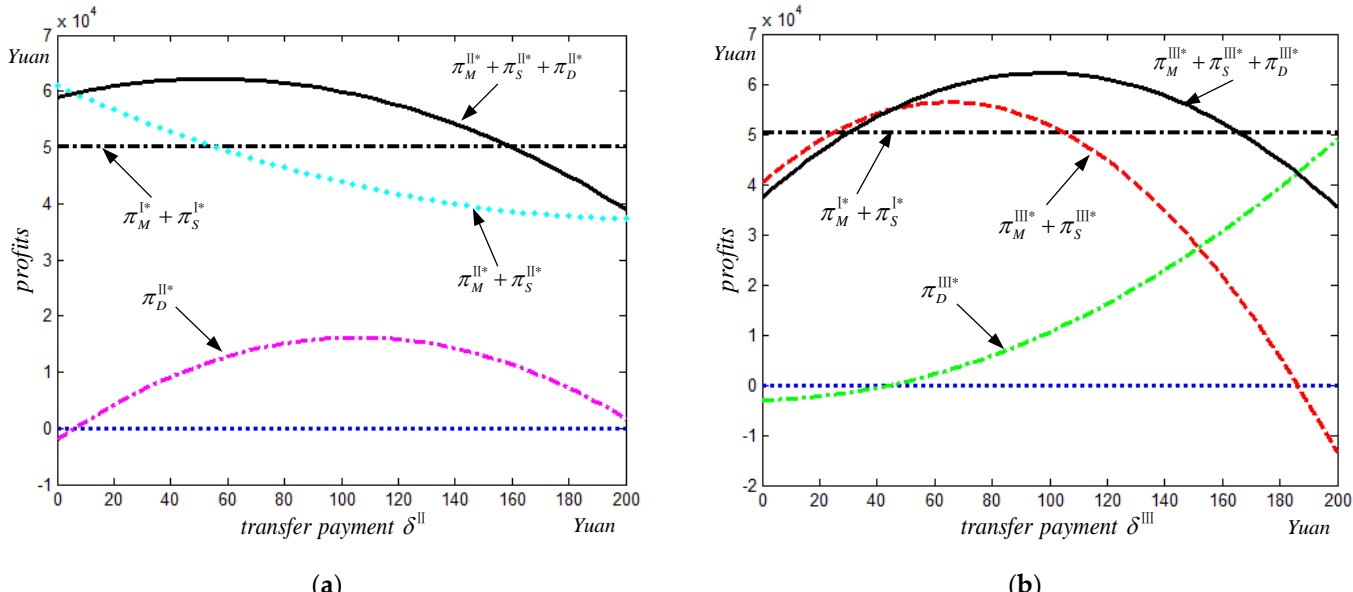

**Figure 6.** (**a**) The effect of transfer payments on the optimal profits of supply chain members in the SMM and SDM models when the supply of remanufactured products is less than the demand; (**b**) the effect of transfer payments on the optimal profits of supply chain members in the SMM and DDM models when the supply of remanufactured products is less than the demand.

### 5.3. Model Selection

Based on the above analysis, we can combine the applicable conditions of all five models and obtain the relationship between transfer payments and optimal overall supply chain profits in different models when the supply of remanufactured products is less than the demand (Figure 8).

Figure 8 shows that all other models in their feasible range can generate a higher overall profit than the SMM model. From the perspective of the whole supply chain, combining the applicable conditions of each model, when the value of the transfer payment $\delta$ is between 6.08 and 12.23, supply chain members would be better off using the SDM model. If the value is between 12.33 and 71.99, supply chain members should use the SDR model. Moreover, if the value is between 71.99 and 115.1, the DDR model is the best option for the supply chain. In other cases, supply chain members may opt for the SMM model. The above analysis shows that the choice of recycling and remanufacturing channels is not absolute. Different channels have their own rules that are affected by transfer payment and various other factors, which will be discussed below. Furthermore, the above analysis results are reciprocal; i.e., the optimal transfer payment range of the specific model can also be obtained through the above analysis.

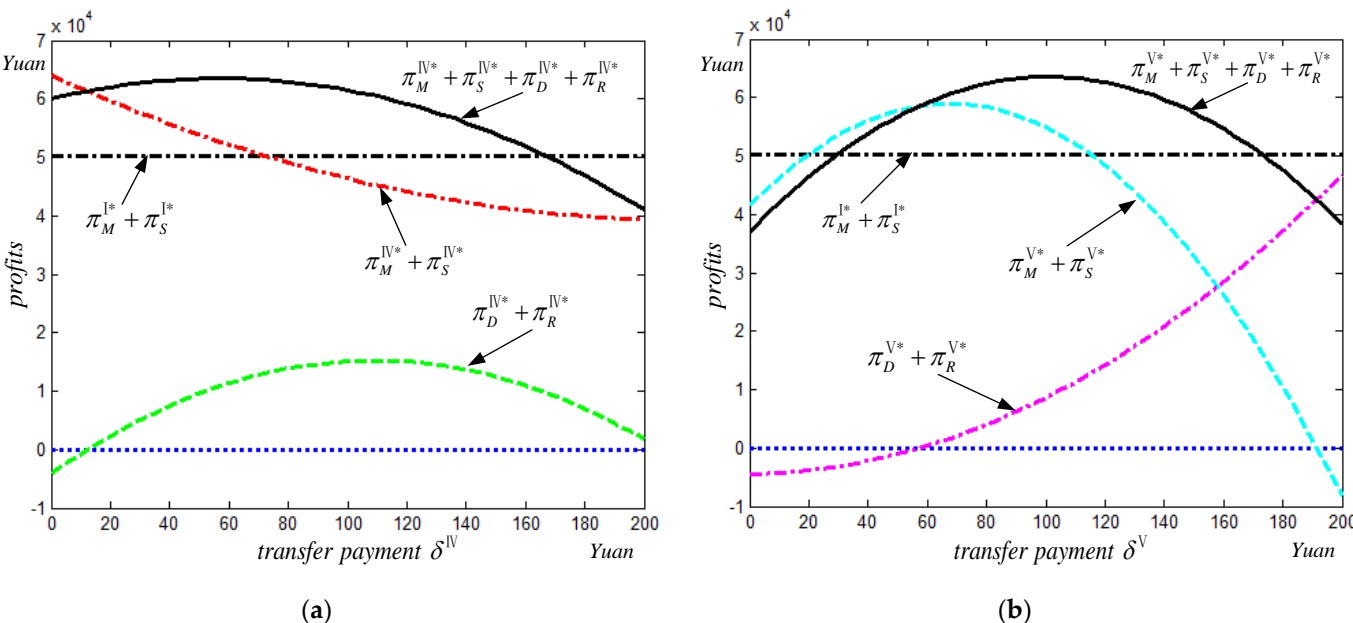

(**a**)                                                                                          (**b**)

**Figure 7.** (**a**) The effect of transfer payments on the optimal profits of supply chain members in the SMM and SDR models when the supply of remanufactured products is less than the demand; (**b**) the effect of transfer payments on the optimal profits of supply chain members in the SMM and DDR models when the supply of remanufactured products is less than the demand.

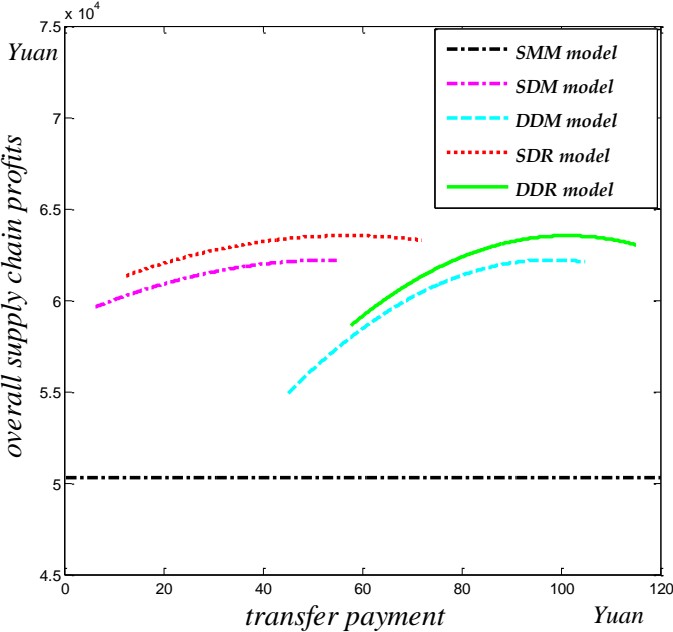

**Figure 8.** The optimal overall profits of the supply chain vary in five models when the supply of remanufactured products is less than the demand.

### 5.4. Influence of Cooperation among Supply Chain

Figures 6 and 7 show that the applicable conditions (transfer payment) when the supply chain as a whole makes the most profit differ from those when each entity makes the most profit. Therefore, whether the supply chain cooperates will affect model selection. This section further investigates the applicable conditions of each model in the case of supply chain cooperation and noncooperation when the supply of remanufactured products is less than demand, as shown in Table 11. In the table, $\alpha$ is the sensitivity coefficient of new products' price to market capacity, which will affect the profits of all entities and the

applicable conditions. The column "transfer payment" indicates the applicable condition of each model in different situations. The column "M + S" shows the range of optimal total profit for the manufacturer and seller, as well as the range of transfer payment extremum when the maximum profit is obtained. Sometimes, the profit rises first, then falls, and the figure between the brackets {} represents the maximum profit in that case. Similarly, the columns "D + R" and "ALL" show the results for the dismantling enterprise and remanufacturer and the entire supply chain, respectively.

By comparing the optimal profits of various scenarios, we can observe that the extremum ranges for the manufacturer and seller are always different from those for the dismantling enterprise and remanufacturer, which makes it difficult for both parties to reach an agreement and achieve maximum profits in a noncooperative scenario, while collaboration between enterprises can promote a CLSC to maximize overall profit in all the models (Table 11). Furthermore, the lower the sensitivity coefficient in the same model, the greater the profit of the supply chain and its members. Therefore, the lower the price elasticity of new products, the more beneficial to the supply chain. Comparing the optimal profits of the SMM, SDM, and DDM models in a noncooperative scenario, we determine that, when there is no remanufacturer, for all parties in the supply chain who seek benefits and make their own decisions, the SDM model can theoretically obtain the highest profits. However, in the noncooperative scenario, the parties' maximum profits cannot be achieved simultaneously, so the parties will bargain over the transfer payment. The SDM and DDM models can achieve the same maximum profits in a cooperative scenario.

In addition, we analyze the situation of the entire CLSC. By comparing the optimal profits of all models in a noncooperative scenario, we found that the SDR model can theoretically achieve the highest profits for all parties in the supply chain who seek benefits and make their own decisions. Similar to the previous analysis, the maximum profits of the parties cannot be achieved at the same time in the noncooperative scenario, so the parties will bargain over the transfer payment. Furthermore, if the manufacturer and seller were aware of the dismantling enterprise's and remanufacturer's private revenue information, they would prefer the DDR model because the divergence between the two parties due to profit differences is smaller. In a cooperative scenario, members' optimal profits in the DDR and SDR models are also higher than in other models. This suggests that more specialized division and cooperation can improve supply chain revenue. Moreover, the cooperative scenario is more conducive to attracting third-party enterprises to participate in the recycling and remanufacturing process. To some extent, it is similar to the value co-creation strategy in logistics industry [55], collaboration between the core enterprises in agricultural machinery CLSC, and the third-party enterprises can mediate the relationship between resource commitment and performance.

**Table 11.** The influence of cooperation among supply chain on model selection when the supply of remanufactured products is less than demand.

| | | Noncooperation * | | | | | | | Cooperation | | |
|---|---|---|---|---|---|---|---|---|---|---|---|
| | | Transfer Payment | M + S | | D + R | | ALL | | Transfer Payment | ALL | |
| | | | Profit Range (×10⁴) | Extremum Range | Profit Range (×10⁴) | Extremum Range | Profit Range (×10⁴) | Extremum Range | | Profit Range (×10⁴) | Extremum Range |
| SMM model | α = 0.8 | / | 7.354 | / | / | / | 7.354 | / | / | 7.354 | / |
| | α = 1 | / | 5.027 | / | / | / | 5.027 | / | / | 5.027 | / |
| | α = 1.2 | / | 3.57 | / | / | / | 3.57 | / | / | 3.57 | / |
| SDM model | α = 0.8 | [4.89, 63.83] | [8.776, 7.354] | 4.89 | [0, 1.852] | 63.83 | [8.776, 9.206] | 63.83 | [0, 197.42] | [8.706, 9.207)<br>{9.207}<br>(9.207, 7.354] | [65.3, 69.72] |
| | α = 1 | [6.08, 54.68] | [5.962, 5.027] | 6.08 | [0, 1.194] | 54.68 | [5.893, 6.22] | [51.73, 54.68] | [0, 158.88] | [5.892, 6.22)<br>{6.22}<br>(6.22, 5.027] | [51.73, 57.48] |
| | α = 1.2 | [7.29, 49] | [4.234, 3.57] | 7.29 | [0, 0.8282] | 49 | [4.234, 4.399)<br>{4.399}<br>(4.399, 4.398] | [44.26, 47.74] | [0, 132.91] | [4.166, 4.399)<br>{4.399}<br>(4.399, 3.57] | [44.26, 47.74] |
| DDM model | α = 0.8 | [44.99, 127.13] | [7.66, 8.161)<br>{8.161}<br>(8.161, 7.354] | [79.62, 82.73] | [0, 0.574)<br>[0.574, 0.6403]<br>(0.6403, 1.847) | 127.13 | [7.66, 9.207)<br>{9.207}<br>(9.207, 9.201] | [121, 123.88] | [0, 207.14] | [7.66, 9.207)<br>{9.207}<br>(9.207, 9.201] | [121, 123.88] |
| | α = 1 | [44.99, 104.93] | [5.489, 5.641)<br>{5.641}<br>(5.641, 5.027] | [64.24, 65.5] | [0, 0.2821)<br>[0.2821, 0.3038]<br>(0.3038, 1.182) | 104.93 | [5.489, 6.22)<br>{6.22}<br>(6.22, 6.209] | [96.32, 100.12] | [30.22, 166.22] | [5.027, 6.22)<br>{6.22}<br>(6.22, 5.027] | [96.32, 100.12] |
| | α = 1.2 | [44.99, 90.3] | [4.044, 4.075)<br>{4.075}<br>(4.075, 3.57] | [53.03, 54.97] | [0, 0.1066)<br>[0.1066, 0.1347]<br>(0.1347, 0.8116) | 90.3 | [4.044, 4.399)<br>{4.399}<br>(4.399, 4.381] | [80.93, 83.2] | [0, 138.77] | [4.044, 4.399)<br>{4.399}<br>(4.399, 4.381] | [80.93, 83.2] |
| SDR model | α = 0.8 | [10.09, 78.4] | [8.965, 7.354] | 10.09 | [0, 1.997] | 78.4 | [8.965, 9.359)<br>{9.359}<br>(9.359, 9.351] | [68.38, 71.49] | [0, 205.04] | [8.821, 9.359)<br>{9.359}<br>(9.359, 7.354] | [68.38, 71.49] |
| | α = 1 | [12.23, 71.99] | [6.132, 5.027] | 12.23 | [0, 1.302] | 71.99 | [6.132, 6.353)<br>{6.353}<br>(6.353, 6.328] | [55.43, 58.63] | [0, 166.9] | [5.995, 6.353)<br>{6.353}<br>(6.353, 5.027] | [55.43, 58.63] |
| | α = 1.2 | [14.46, 69.32] | [4.391, 3.57] | 14.46 | [0, 0.9007] | 69.32 | [4.391, 4.518)<br>{4.518}<br>(4.518, 4.475] | [45.72, 51.13] | [0, 141.39] | [4.261, 4.518)<br>{4.518}<br>(4.518, 3.57] | [45.72, 51.13] |
| DDR model | α = 0.8 | [57.6, 136.86] | [8.167, 8.43)<br>{8.43}<br>(8.43, 7.354] | [82.7, 84.94] | [0, 0.4553)<br>[0.4553, 0.5035]<br>(0.5035, 1.972) | 136.86 | [8.167, 9.359)<br>{9.359}<br>(9.359, 9.326] | [124.57, 126.6] | [37.44, 213.68] | [7.354, 9.359)<br>{9.359}<br>(9.359, 7.354] | [124.57, 126.6] |
| | α = 1 | [57.6, 115.18] | [5.859, 5.896)<br>{5.896}<br>(5.896, 5.027] | [66.2, 68.83] | [0, 0.1382)<br>[0.1382, 0.1842]<br>(0.1842, 0.277] | 115.18 | [5.027, 6.353)<br>{6.353}<br>(6.353, 5.027] | [100.35, 102.43] | [0, 173.03] | [5.027, 6.353)<br>{6.353}<br>(6.353, 5.027] | [100.35, 102.43] |
| | α = 1.2 | [57.6, 100.99] | [4.321, 3.57] | 57.6 | [0, 0.8849] | 100.99 | [4.321, 4.518)<br>{4.518}<br>(4.518, 4.454] | [83.45, 86.97] | [24.61, 145.85] | [3.57, 4.518)<br>{4.518}<br>(4.518, 3.57] | [83.45, 86.97] |

* The unit of data is Yuan.

*5.5. Influence of Recovery and Remanufacturing Prices*

The price paid by a manufacturer and seller to third-party enterprises for recovery or remanufacturing is the major component of transfer payment. In this section, we examine the impacts of the recovery price of available machine parts sold by the dismantling enterprise to the manufacturer $p_{r,DM}$ and the wholesale price of remanufactured products produced by the remanufacturer $w_{Rr}$ on optimal profits and channel selection when the supply of remanufactured products is less than the demand. To simplify calculations and analyses, we assume that the recovery price of scrapped products sold by seller to dismantling enterprise is 1.1 times the recovery price of scrapped products sold by consumer to seller, that is, $p_{r,SD} = 1.1p_{r,S}$.

As shown in Figure 9, when there is no remanufacturer, the manufacturer's recovery price $p_{r,DM}$ will have a completely different influence on the optimal profits of all parties in the SDM and DDM models. According to the applicable condition, the SDM model will exist only when the recovery price is relatively high. As the recovery price rises, the optimal profit of the dismantling enterprise increases first and then decreases in the SDM model. Meanwhile, the optimal profit of a dismantling enterprise is always higher in the DDM model than in the SDM model and grows exponentially. Furthermore, an increase in the recovery price reduces the optimal total profit of the manufacturer and seller in the SDM model, but the total profit remains higher than in the DDM model. In the DDM model, an increase in the recovery price encourages the recovery of scrapped products, increasing the output of remanufactured products, and thus the optimal profit of the manufacturer and seller increases first and then decreases. To summarize, when the recovery price of available machine parts sold by a dismantling enterprise to a manufacturer is relatively low, the supply chain should use the DDM model. The manufacturer can increase the recovery price appropriately to improve the overall supply chain performance. However, as the recovery price rises further, the manufacturer and seller will gradually shift to the SDM model, whereas the dismantling enterprise will continue to prefer the DDM model, resulting in a divergence between the two sides.

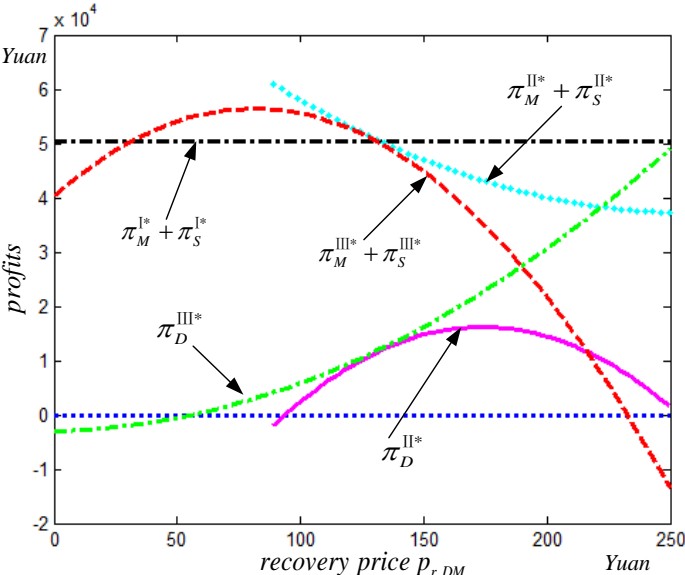

**Figure 9.** The influence of manufacturer's recovery price on supply chain members' optimal profits when the supply of remanufactured products is less than demand.

Similar to the previous analysis, when remanufacturers are present in the supply chain, as shown in Figure 10, the remanufacturer's wholesale price $w_{Rr}$ has a completely different influence on the optimal profits of all parties in the SDR model and the DDR model. Due to the applicable condition, the SDR model will exist only when the wholesale price is relatively high. As the wholesale price rises, the optimal total profit of the dismantling

enterprise and remanufacturer rises first, then falls in the SDR model. In the DDR model, the optimal total profit of the dismantling enterprise and remanufacturer is always greater than in the SDR model and grows exponentially. Furthermore, an increase in wholesale price reduces the optimal total profit of the manufacturer and seller in the SDR model, but the total profit remains higher than in the DDR model. In the DDR model, an increase in wholesale price promotes output of remanufactured products, causing the optimal total profit of the manufacturer and seller to rise first and then fall. To summarize, when the wholesale price of remanufactured products produced by a remanufacturer is relatively low, the supply chain should use the DDR model, and remanufacturers can appropriately raise the wholesale price to improve the overall supply chain performance. However, as wholesale prices rise further, manufacturers and sellers will gradually shift to the SDR model, while dismantling enterprises and remanufacturers will continue to prefer the DDR model, resulting in a schism between the two sides.

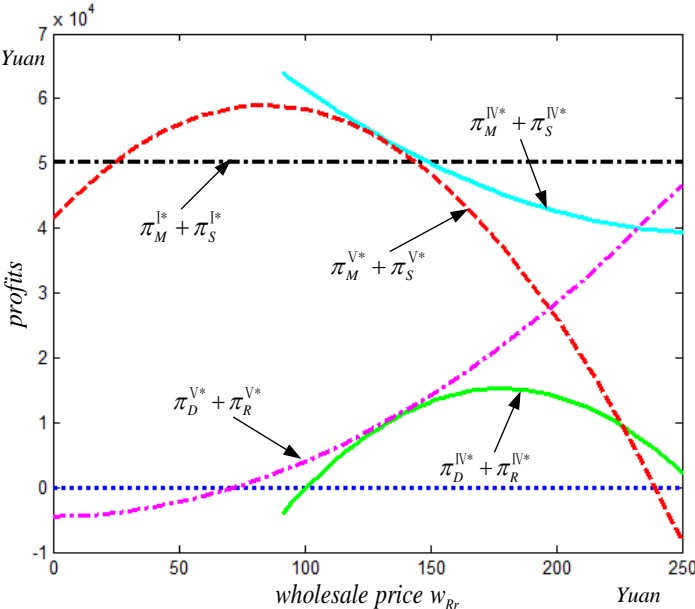

**Figure 10.** The influence of remanufacturer's wholesale price on supply chain members' optimal profits when the supply of remanufactured products is less than demand.

*5.6. A Special Case: Oversupply*

In the previous section, the research focused on a situation in which the supply of remanufactured products was less than the demand, and there was occasionally an oversupply of remanufactured products. Thus, in this section, we consider the case of oversupply and compare recycling of scrapped products in the models above. Figure 11 depicts the optimal recovery price of scrapped products in each model when supply exceeds demand for remanufactured products.

As shown in Figure 11, even if the influence coefficient of recovery price of scrapped products on recovery quantity $\mu$ is very large, which means, although the recovery price can significantly increase the recovery quantity, the optimal recovery prices of scrapped products sold by consumers in the five recycling models are all less than zero. That is, when the supply of remanufactured products exceeds the demand, the enterprises that recover the scrapped products will suffer losses. As a result, a CLSC faces risks when recycling and remanufacturing, and it should be slowed or stopped when the supply of remanufactured products exceeds demand. If the government wants to encourage enterprises to recycle and remanufacture all the time, it should provide certain subsidies [56] to enterprises when supply exceeds demand in order to help the CLSC overcome difficulties.

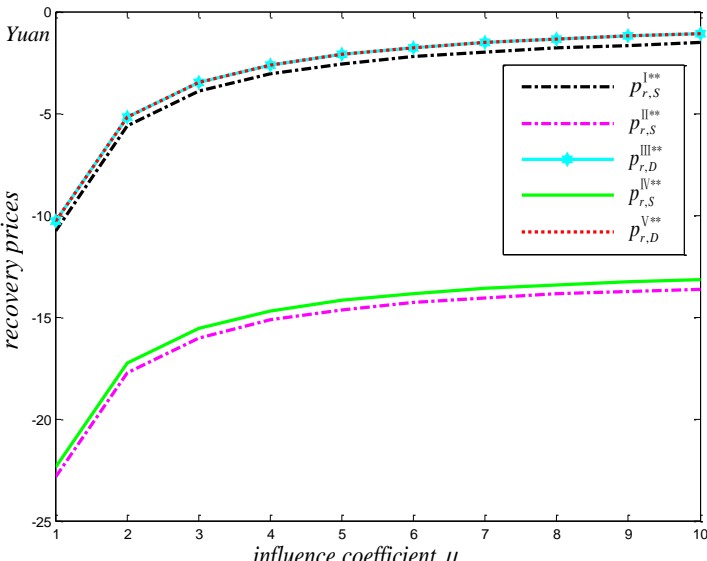

**Figure 11.** The optimal recovery prices of scrapped products sold by a consumer in different models when the supply of remanufactured products is more than demand.

## 6. Conclusions

Motivated by the practice in China's agricultural machinery remanufacturing industry, we focus on the CLSC of agricultural machinery and present five models for recycling and remanufacturing channels. The pricing decisions and corresponding profits of CLSC members are analyzed in the five models. Especially, we study the conditions that make selection of recycling channels more beneficial to members of the CLSC when there is a price difference between new and remanufactured products. We then use numerical examples to demonstrate the applicable condition of each model and the basis for CLSC members' channel selection. On this basis, the impact of various factors, such as supply chain cooperation, recovery and remanufacturing prices, and supply and demand for remanufactured products, have been thoroughly studied.

This study presents a theoretical analysis framework and numerical calculation method for the channel selection of a CLSC for remanufacturing scrapped agricultural machines. It can provide members of the agricultural machinery remanufacturing supply chain in China with channel selection guidance, which will help in solving the current problem of chaotic and changeable remanufacturing channels caused by inaccurate decision-making and also provide references for other countries with similar experiences as China in agricultural machinery remanufacturing. It also helps to promote progress regarding selection of recycling channels and differentiated pricing strategies research and development of the field of agricultural machinery remanufacturing management and provides a novel direction for subsequent related research.

The research results indicate that no one recycling and remanufacturing channel is clearly superior to the others. Different channels have their own rules that are affected by transfer payments and various other factors. To be specific, (i) as transfer payments paid by a manufacturer and seller to a third-party enterprise increase, the supply chain should select SMM, SDM, SDR, and DDR channels contrapuntally to achieve a win–win situation for all members, and members in a specific channel should adjust the transfer payment according to optimal payment range. (ii) When there is no remanufacturer, the supply chain may adopt the SDM channel, but maximum profits of the parties cannot be achieved simultaneously in a noncooperative scenario. Meanwhile, in a cooperative scenario, SDM and DDM channels can achieve the same maximum profits. (iii) When there are remanufacturers in the supply chain, in a noncooperative scenario, a manufacturer and seller would choose the DDR channel if they knew the private revenue information of the dismantling enterprise and remanufacturer; otherwise, they would prefer the SDR channel.

In the cooperative scenario, the performance of DDR and SDR channels is better than other channels. (iv) When the recovery price of available machine parts sold by a dismantling enterprise to a manufacturer is relatively low, the supply chain should choose the DDM channel and the manufacturer can appropriately increase the recovery price to improve the performance of the entire supply chain; however, as the recovery price increases further, the manufacturer and seller will turn to the SDM channel, while the dismantling enterprise will still prefer the DDM channel. (v) When the wholesale price of remanufactured products produced by the remanufacturer is relatively low, the supply chain should select the DDR channel. Moreover, remanufacturers can appropriately increase the wholesale price so as to improve the performance of the whole supply chain, but, with the further increase in wholesale price, the manufacturer and seller would turn to the SDR channel, whereas the dismantling enterprise and remanufacturer would still prefer the DDR channel. (vi) When the supply of remanufactured products is less than the demand, the optimal selling prices of new products are the same across all channels, whereas, when supply exceeds demand, the optimal recovery price of scrapped products sold by consumers is negative in all five channels, which will lead to disruptions in recycling and remanufacturing.

From the main findings in this paper, we can conclude some managerial insights for the government and enterprises in the agricultural machinery remanufacturing industry. First, a cooperative scenario that promotes a CLSC to maximize overall profit is more conducive to attracting third-party enterprises to participate in the remanufacturing process to form more specialized divisions and increase supply chain revenue. Thus, for the members of the CLSC, with the increasing number of agricultural machines that need to be scrapped, some strategies should be used appropriately on channel selection and cooperate with each other to meet the conditions, a win–win situation both parties can achieve. For new entrants in the industry, more open and inclusive channels should be chosen rather than those that exclude third-party enterprises. In addition, the agricultural machinery recycling and remanufacturing industry in China is mainly composed of small- and medium-sized enterprises [5]. In order to maintain and strengthen supply chain cooperation scenarios, a community of practice can be established to create a strong network among supply chain members and eliminate the informative gap [57], and, relying on the government's strong policy enforcement, the Chinese government should introduce effective policies that encourage collaboration and information/knowledge exchange between key actors [58,59] to promote agricultural machinery recycling and remanufacturing. Second, in the same recycling channel, the lower the price elasticity of new products, the more beneficial to the supply chain, so recycling key machine parts rather than other replaceable parts is more beneficial for the enterprises. However, in the meantime, recycling of noncritical and unimportant machine parts should be well considered by the government to avoid environmental pollution and social problems. Furthermore, when remanufactured products oversupply, recovery of scrapped products becomes unprofitable and government subsidies are required to maintain sustainable recycling and remanufacturing of scrapped agricultural machines. Specifically, some targeted policies can be created in China to promote sustainable agricultural machinery recycling, such as recovery price subsidies [56], tax exemptions [60], and funding research activities [58].

Our study also has some limitations related to the model and methodology used. Multiple recycling and remanufacturing channels may exist and compete at the same time, and there may be competition among peer enterprises in the CLSC. Therefore, this suggests numerous avenues for future research, including considering multiple competitive recycling channels and competition among multiple peer enterprises in the supply chain. Furthermore, the government's strategy for subsidies on agricultural machinery remanufacturing and channel selection of the CLSC under government subsidies are both worthy of attention.

**Supplementary Materials:** The following supporting information can be downloaded at: https://www.mdpi.com/article/10.3390/su15065337/s1; As an example, the MATLAB (Version R2010a) code and dataset for applicable conditions of the SMM and SDM models. Readers can calculate and reproduce the results in numerical analysis by adjusting some functions and parameter settings.

**Author Contributions:** Conceptualization, G.Z. and J.C.; data curation, L.Z. and H.L.; formal analysis, L.Z. and H.L.; funding acquisition, L.Z.; investigation, L.Z. and G.Z.; methodology, J.C.; project administration, L.Z.; resources, G.Z.; software, H.L.; supervision, G.Z.; validation, L.Z.; visualization, H.L.; writing—original draft preparation, L.Z. and H.L.; writing—review and editing, L.Z., G.Z. and J.C. All authors have read and agreed to the published version of the manuscript.

**Funding:** This research was funded by the Science Foundation of Ministry of Education of China (grant number 20YJC630235) and the Zhejiang Provincial Natural Science Foundation of China (grant number LQ20G020008).

**Institutional Review Board Statement:** Not applicable.

**Informed Consent Statement:** Not applicable.

**Data Availability Statement:** Data are contained within the article.

**Acknowledgments:** The authors would like to thank the local agricultural management departments in Hangzhou, Shaoxing, and Ningbo of Zhejiang Province for their help, the enterprises under investigation for their cooperation, and the local agricultural machinery consumers for their participation. The authors would also like to thank the editors and anonymous reviewers for their review and guidance.

**Conflicts of Interest:** The authors declare no conflict of interest.

## Appendix A

**Proof of Proposition 1.** In the SMM model, because of $q = -\alpha p_n + B$, $c_{Mr} = c_{Mn} - \eta_M \theta_M$, and $Q_r = A + \mu p_{r,S} + \varepsilon$, and these can be substituted into Equation (2) and obtain that:

$$
\begin{aligned}
\pi_M^I + \pi_S^I &= (p_n - c_{Mn})q + [(p_r - c_{Mr}) \cdot \theta_M - (p_{r,S} + c_{r,M})]Q_r - C_{M1} - C_{M2} - C_{M3} - C_{S1} - C_{S2} \\
&= (p_n - c_{Mn}) \cdot (-\alpha p_n + B) \\
&\quad + [(p_r - c_{Mn} + \eta_M \theta_M) \cdot \theta_M - (p_{r,S} + c_{r,M})] \cdot (A + \mu p_{r,S} + \varepsilon) \\
&\quad - C_{M1} - C_{M2} - C_{M3} - C_{S1} - C_{S2}
\end{aligned}
\tag{A1}
$$

and we can obtain that:

$$
\frac{\partial(\pi_M^I + \pi_S^I)}{\partial p_n} = -2\alpha p_n + B + \alpha c_{Mn}
\tag{A2}
$$

$$
\frac{\partial(\pi_M^I + \pi_S^I)}{\partial p_{r,S}} = -2\mu p_{r,S} + \mu[(p_r - c_{Mn} + \eta_M \theta_M) \cdot \theta_M - c_{r,M}] - (A + \varepsilon)
\tag{A3}
$$

$$
\frac{\partial(\pi_M^I + \pi_S^I)}{\partial p_r} = \theta_M(A + \mu p_{r,S} + \varepsilon) > 0
\tag{A4}
$$

To solve this problem, we first consider $p_r$ as a constant and obtain the Hessian matrix of Equation (A1) as follows:

$$
H_1 = \begin{bmatrix} \frac{\partial^2(\pi_M^I + \pi_S^I)}{\partial p_n^2} & \frac{\partial^2(\pi_M^I + \pi_S^I)}{\partial p_n \partial p_{r,S}} \\ \frac{\partial^2(\pi_M^I + \pi_S^I)}{\partial p_{r,S} \partial p_n} & \frac{\partial^2(\pi_M^I + \pi_S^I)}{\partial p_{r,S}^2} \end{bmatrix} = \begin{bmatrix} -2\alpha & 0 \\ 0 & -2\mu \end{bmatrix}
\tag{A5}
$$

The Hessian matrix is then negative definite, and, combining $\frac{\partial(\pi_M^I + \pi_S^I)}{\partial p_n} = 0$, $\frac{\partial(\pi_M^I + \pi_S^I)}{\partial p_{r,S}} = 0$, we can obtain that:

$$
p_n^{I*} = \frac{B + \alpha c_{Mn}}{2\alpha}
\tag{A6}
$$

$$p_{r,S}^{I*} = \frac{1}{2}[(p_r - c_{Mn} + \eta_M\theta_M) \cdot \theta_M - c_{r,M}] - \frac{A+\varepsilon}{2\mu} \tag{A7}$$

In addition, as $\frac{\partial(\pi_M^I + \pi_S^I)}{\partial p_r} > 0$, the optimal $p_r$ would be close to $p_n$ in reality, and then $q_r = \beta(p_n - p_r)$ would also be small enough, and, combining with the condition $q_r \geq \theta_M Q_r$, we can obtain that:

$$\beta(p_n - p_r) = \theta_M(A + \mu p_{r,S} + \varepsilon) \tag{A8}$$

Combining Equations (A7) and (A8), we have the following:

$$p_r^{I*} = \frac{2\beta p_n^{I*} - (A+\varepsilon)\theta_M + \mu\theta_M[(c_{Mn} - \eta_M\theta_M) \cdot \theta_M + c_{r,M}]}{2\beta + \mu\theta_M^2} \tag{A9}$$

$$p_{r,S}^{I*} = \frac{\beta(p_n^{I*} - p_r^{I*})}{\mu\theta_M} - \frac{A+\varepsilon}{\mu} \tag{A10}$$

Thereby, the optimal solution is indicated in Equations (A6), (A9) and (A10). □

**Proof of Proposition 2.** In the SMM model, because of $q = -\alpha p_n + B$, $c_{Mr} = c_{Mn} - \eta_M\theta_M$, $q_r = \beta(p_n - p_r)$, and $Q_r = A + \mu p_{r,S} + \varepsilon$, and these can be substituted into Equation (6) to obtain that:

$$\begin{aligned}\pi_M^I + \pi_S^I &= (p_n - c_{Mn}) \cdot (-\alpha p_n + B) + (p_r - c_{Mn} + \eta_M\theta_M + c_{s,S}) \cdot \beta(p_n - p_r) \\ &\quad -(p_{r,S} + c_{r,M} + c_{s,S}\theta_M)(A + \mu p_{r,S} + \varepsilon) - C_{M1} - C_{M2} - C_{M3} - C_{S1} - C_{S2}\end{aligned} \tag{A11}$$

and we can obtain that:

$$\frac{\partial(\pi_M^I + \pi_S^I)}{\partial p_n} = -2\alpha p_n + \beta p_r + B + \alpha c_{Mn} + \beta(\eta_M\theta_M + c_{s,S} - c_{Mn}) \tag{A12}$$

$$\frac{\partial(\pi_M^I + \pi_S^I)}{\partial p_r} = \beta(p_n - p_r) - \beta(p_r - c_{Mn} + \eta_M\theta_M + c_{s,S}) \tag{A13}$$

$$\frac{\partial(\pi_M^I + \pi_S^I)}{\partial p_{r,S}} = -(A + \mu p_{r,S} + \varepsilon) - \mu(p_{r,S} + c_{r,M} + c_{s,S}\theta_M) \tag{A14}$$

Therefore, we can obtain the Hessian matrix of Equation (A11) as follows:

$$H_1 = \begin{bmatrix} \frac{\partial^2(\pi_M^I + \pi_S^I)}{\partial p_n^2} & \frac{\partial^2(\pi_M^I + \pi_S^I)}{\partial p_n \partial p_r} & \frac{\partial^2(\pi_M^I + \pi_S^I)}{\partial p_n \partial p_{r,S}} \\ \frac{\partial^2(\pi_M^I + \pi_S^I)}{\partial p_r \partial p_n} & \frac{\partial^2(\pi_M^I + \pi_S^I)}{\partial p_r^2} & \frac{\partial^2(\pi_M^I + \pi_S^I)}{\partial p_r \partial p_{r,S}} \\ \frac{\partial^2(\pi_M^I + \pi_S^I)}{\partial p_{r,S} \partial p_n} & \frac{\partial^2(\pi_M^I + \pi_S^I)}{\partial p_{r,S} \partial p_r} & \frac{\partial^2(\pi_M^I + \pi_S^I)}{\partial p_{r,S}^2} \end{bmatrix} = \begin{bmatrix} -2\alpha & \beta & 0 \\ \beta & -2\beta & 0 \\ 0 & 0 & -2\mu \end{bmatrix} \tag{A15}$$

The Hessian matrix is then negative definite, and, combining $\frac{\partial(\pi_M^I + \pi_S^I)}{\partial p_n} = 0$, $\frac{\partial(\pi_M^I + \pi_S^I)}{\partial p_r} = 0$, $\frac{\partial(\pi_M^I + \pi_S^I)}{\partial p_{r,S}} = 0$, we can obtain that:

$$p_n^{I**} = \frac{2B + 2\alpha c_{Mn} + \beta(\eta_M\theta_M + c_{s,S} - c_{Mn})}{4\alpha - \beta} \tag{A16}$$

$$p_r^{I**} = \frac{B + \alpha c_{Mn} - (2\alpha - \beta)(\eta_M\theta_M + c_{s,S} - c_{Mn})}{4\alpha - \beta} \tag{A17}$$

$$p_{r,S}^{I**} = -\frac{c_{r,M} + c_{s,S}\theta_M}{2} - \frac{A+\varepsilon}{2\mu} \tag{A18}$$

Thereby, the optimal solution is indicated in Equations (A16)–(A18). □

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
