# Peer review of "Channel Selection of Closed-Loop Supply Chain for Scrapped Agricultural Machines Remanufacturing"

_sustainability, doi:10.3390/su15065337_

Round 1

Reviewer 1 Report

This article develops and optimizes mathematical models of several closed-loop supply chain options for agricultural machinery in China.

I commend the authors for stating so clearly that "no single recycling channel is definitely superior to others" - so much work goes into attempting to identify one best way to increase or implement circularity that it's refreshing to see a more inconclusive conclusion, and I believe this conclusion to be much more reflective of real-world conditions than typical optimization outcomes.

Overall, I'd like to see some revisions to this article before it is published. My main concerns are as follows:

1. The model results are dependent on six assumptions listed in lines 246 - 259, and some of these assumptions are very strong and/or unlikely to be true in reality. In particular, assumption 1 is obviously necessary if a neat mathematical model is to be formulated and solved, but the assumption is very restrictive. There is a little discussion towards the end of this article on the fact that there are assumptions which may not align with the real world. I would like to see more thought from the authors on how the mathematical results in this paper can be used by real-world stakeholders and decision makers - perhaps not to develop or implement specific CLSC channels, but more generally in deciding how to participate in such channels or establish new enterprises to participate in such channels. 

2. The motivation and scientific need for this type of modeling work is not presented clearly, and I'm not convinced that agricultural machinery circularity in China needs an entirely new research article dedicated to the topic. I found the literature review to be lengthy but directionless: it was primarily a list of references and topics without a lot of discussion on how the present work relates to the extant work. In particular, I didn't feel after reading the literature review that there was a significant knowledge gap that needed to be filled. Are the authors trying to make the case that CLSCs for agricultural machinery are sufficiently distinct from CLSCs for other large machinery types that a separate modeling and analysis effort is needed? (Lines 113-116 seem to indicate so.) In line 217, the model description is begun with two references (4 and 5), neither of which involve agricultural machinery, and yet it seems like those references are the basis for the models which follow. My recommendation is to rewrite the literature review so it's clear what extant literature does and does not do and use that to clearly define the motivation and need for the present work. In particular, answer the question of why ag machinery is so unique that it requires new modeling and analysis work.

3. All of the mathematical proofs - which serve as the basis for the numerical results and conclusions - are very terse and would require non-trivial effort on the part of the reader to verify. The authors need to write out more steps, in effect "showing their work", perhaps in a Supporting Information or Appendix section. Because all of the proofs are very similar, I think doing this for one or two representative proofs would be sufficient.

Some smaller and more specific comments are as follows:

- Lines 257-258: It's not clear to me how assumption 6 affects the model or the results. Can the authors add some explanation?

- Line 94: Seems like a strong and inaccurate statement that "Each enterprise makes blind decisions". Surely enterprises are already making what they perceive to be profit-maximizing decisions, or at least decisions that they perceive in some way to be beneficial?

- Line 95: "significant decrease in farmers' enthusiams" What is the farmer incentive to participate in ag machinery CLSCs? Are they required to by the policies in Table 1?

- Line 276 and following: "recycles" I'd recommend avoiding "recycles" when "collects" or "returns" is meant. "Recycling" has a very specific connotation that is not correct in this context.

- Line 283 and following: While I was reading through the models it wasn't clear to me if transportation costs are included.

- Line 327: I have missed something in the model description, because why are all of the prices determined jointly by manufacturer and seller when not all of the prices apply to both enterprises?

- Figure 8: The x-axis range of this graph is much, much larger than the range of the data being graphed and discussed. I recommend re-drawing this figure so the data fills more of the available space.

- All figures and related discussion: Virtually every figure is missing a y-axis label, which is needed even if the y-axis is described in the caption or in the accompanying text. All axis labels should include units, again even if the units are given elsewhere (which I do not believe they are). Currency units are especially important for a country-specific analysis like this one. The figure text (not the CLSC diagrams, but the graphs specifically) is also quite small throughout and in general the graphs could be more readable.

- Lines 549-551: This is a long, long list of parameters and numbers that have no units and aren't defined in the surrounding discussion. Instead of this list, I recommend a table with all parameters clearly defined and all units listed.

- Line 606: Are "opportunity costs" included explicitly in the model, or is the phrase being used more generally? 

- Lines 679-680: This is a really exciting conclusion! I recommend emphasizing this finding much more heavily, as it's an insight that could be really impactful in the developing of CLSCs.

- In the beginning of the paper, there is much emphasis on the fact that this work focuses on Chinese CLSCs and how provinces in China are required to deal with end-of-life machinery. However, later on in the paper the results and conclusions are worded much more generally. I recommend the authors re-iterate the China-only nature of these insights - or, alternatively, state up-front that although the present work focuses on China, the resulting insights are more broadly applicable (also explain why).

- A note on transparency: The authors might consider publishing their calculations (whether as a codebase, spreadsheet, or other format), input datasets, and numerical results along with the article, provided there are no concerns about proprietary data. It's certainly not a requirement for the article to be published but it's good practice for math-heavy research like this.

Author Response

Response to Reviewer 1 Comments

Point 1: I commend the authors for stating so clearly that "no single recycling channel is definitely superior to others" - so much work goes into attempting to identify one best way to increase or implement circularity that it's refreshing to see a more inconclusive conclusion, and I believe this conclusion to be much more reflective of real-world conditions than typical optimization outcomes.

Response 1: Dear reviewer, firstly, thank you very much for your commendation, which is a very rewarding recognition to our efforts. We are particularly grateful for your valuable and insightful suggestions and comments, which are very important for us to improve the quality of this paper, and should deserve our full respect. According to the constructive suggestions from you and other reviewers, we have devoted a great deal of time and effort to revise this paper. The Literature review section and Conclusions section have been greatly improved, and we have supplemented the references, proofs of derivation process and the example of Matlab code and dataset to further enhance the quality of this paper. And we have also carefully edited the manuscript to eliminate other shortcomings. The revisions to the manuscript have been marked up by using the “Track Changes” function of MS Word. We appreciate all the comments and suggestions, and hope that our revisions match all the requirements. If you have any question about this paper, please don’t hesitate to let me know. Your acknowledgement will be highly appreciated. We will appreciate to receive further comments and suggestions about this paper and actively make the revision in order to enhance the quality of this paper.

Point 2: The model results are dependent on six assumptions listed in lines 246 - 259, and some of these assumptions are very strong and/or unlikely to be true in reality. In particular, assumption 1 is obviously necessary if a neat mathematical model is to be formulated and solved, but the assumption is very restrictive. There is a little discussion towards the end of this article on the fact that there are assumptions which may not align with the real world. I would like to see more thought from the authors on how the mathematical results in this paper can be used by real-world stakeholders and decision makers - perhaps not to develop or implement specific CLSC channels, but more generally in deciding how to participate in such channels or establish new enterprises to participate in such channels.

Response 2: Thank you very much for your constructive suggestions that do support our work. We very much agree with you, and we have revised and enhanced the related content to elaborate the managerial insights of the mathematical results as specific as possible in this paper. Firstly, as you recommended, we focused on the benefits of supply chain collaboration and explored the possible causes in combination with the literature. To some extent, it is similar to the value co-creation strategy in logistics industry, collaboration between the core enterprises in agricultural machinery CLSC and the third-party enterprises can mediate the relationship between resource commitment and performance. Secondly, based on the above, we presented our recommendations for the Chinese government and the enterprises in agricultural machinery remanufacturing industry. Specifically, for new entrants in such industry, more open and inclusive channels should be chosen rather than those that exclude third-party enterprises. In addition, the agricultural machinery recycling and remanufacturing industry in China is mainly composed of small and medium-sized enterprises. In order to maintain and strengthen supply chain cooperation scenarios, the Community of Practice can be established to create strong network among supply chain members and eliminate the informative gap. And relying on the government's strong policy enforcement, the Chinese government should introduce effective policies that encourage collaboration and information/knowledge exchange between key actors to promote the agricultural machinery recycling and remanufacturing. Further, we discussed the government policies when the supply of remanufactured products exceeded demand, and suggested that some targeted policies can be created in China including recovery price subsidies, tax exemptions and funding research activities based on relevant literature. The above content has been supplemented to the discussion of results (lines 697-700) and conclusions (lines 824-850) to make this paper more realistic. Thank you very much for guiding us.

Point 3: The motivation and scientific need for this type of modeling work is not presented clearly, and I'm not convinced that agricultural machinery circularity in China needs an entirely new research article dedicated to the topic. I found the literature review to be lengthy but directionless: it was primarily a list of references and topics without a lot of discussion on how the present work relates to the extant work. In particular, I didn't feel after reading the literature review that there was a significant knowledge gap that needed to be filled. Are the authors trying to make the case that CLSCs for agricultural machinery are sufficiently distinct from CLSCs for other large machinery types that a separate modeling and analysis effort is needed? (Lines 113-116 seem to indicate so.) In line 217, the model description is begun with two references (4 and 5), neither of which involve agricultural machinery, and yet it seems like those references are the basis for the models which follow. My recommendation is to rewrite the literature review so it's clear what extant literature does and does not do and use that to clearly define the motivation and need for the present work. In particular, answer the question of why ag machinery is so unique that it requires new modeling and analysis work.

Response 3: Thank you for your suggestion, and you have accurately pointed out the key problems of this paper. Based on your insightful suggestions and comments, we apologize that our original literature review section is not very clear and not well organized; and it has been rewrited to explain the research progress in related fields and the research gap that currently exists as specific as possible. Firstly, we have reviewed the literature on selection of recycling channels in CLSC on lines 119-158. The recycling channel is a combination of recovery, dismantling, and remanufac-turing processes, which has attracted the attention of many researchers. However, the existing research mainly focuses on the selection of recovery channels. Some scholars have conducted research on the selection of remanufacturing channels, but the current research on channel selection for the entire process of the CLSC including recovery, dismantling and remanufacturing is still limited. Since the structure of agricultural machinery is relatively simpler than that of large construction machinery and automobiles, the recovery, dismantling and remanufac-turing channels of agricultural machinery also involve more third-party enterprises, which is evident in China. Therefore, it is of great practical significance to study the problem of channel selection for the whole process of CLSC. Secondly, the literature on differentiated pricing strategies for remanufactured products are reviewed on lines 159-207. Based on the review, the existing research on differentiated pricing strategies is mainly based on manufacturers, consumer demand, supply chain partners and the external environment, while the research on the pricing strategies of new and remanu-factured products based on different recycling and remanufacturing channels is still limited. Furthermore, research is lacking on the influence of differentiated pricing strategies on the recycling channel selection in CLSC. Therefore, we propose the intention of this paper to make contribution to this area as mentioned in the summary part of literature review section on lines 208-235.

Point 4: All of the mathematical proofs - which serve as the basis for the numerical results and conclusions - are very terse and would require non-trivial effort on the part of the reader to verify. The authors need to write out more steps, in effect "showing their work", perhaps in a Supporting Information or Appendix section. Because all of the proofs are very similar, I think doing this for one or two representative proofs would be sufficient.

Response 4: We do apologize that our original mathematical proofs are oversimplified, and the Proof of Proposition 1 and Proof of Proposition 2 have been revised and enhanced as the representative proofs in Appendix A. In the meantime, thank you very much for your criticism, we did find deficiencies in the original description of proof process when we reviewed it, and now the derivation process has been carefully supplemented and improved. Thank you very much for guiding us.

Point 5: Lines 257-258: It's not clear to me how assumption 6 affects the model or the results. Can the authors add some explanation?

Response 5: We apologize that our original related statement in the model development part is not clear, and it has already been revised in this paper. Assumption 6 actually explains the relationship between the remanufacturer and dismantling enterprise in the models. When there is a remanufacturer in the CLSC, the dismantling enterprise is usually an affiliate of the remanufacturer. Therefore, in the DDR model, they are both a community of interests and make decisions together. And when there is only dismantling enterprise in the CLSC, it makes decisions independently, as represented in the DDM model. We have revised the related content to explain it more clearly on line 390 and line 489. Thank you very much for reminding us.

Point 6: Line 94: Seems like a strong and inaccurate statement that "Each enterprise makes blind decisions". Surely enterprises are already making what they perceive to be profit-maximizing decisions, or at least decisions that they perceive in some way to be beneficial?

Response 6: We apologize that our original statement “Each enterprise makes blind decisions” is inaccurate, as you pointed out, and it has been replaced by “Each enterprise makes short-term decisions” in this paper. It means that the relevant enterprises usually make decisions based on relatively partial information and speculation. As a result, farmers' desire to use policies to update their agricultural machinery has been hindered. Also the original statement “blind decision-making” on line 789 has been replaced by “inaccurate decision-making”. Thank you very much.

Point 7: Line 95: "significant decrease in farmers' enthusiams" What is the farmer incentive to participate in ag machinery CLSCs? Are they required to by the policies in Table 1?

Response 7: We apologize that our original statement about “significant decrease in farmers' enthusiams” is not clear, and it has been replaced by “a serious obstacle to the desire of farmers to update agricultural machines” in this paper. The policies in Table 1 encourage farmers to upgrade their machinery, and farmers can not only get new machinery (at a reasonable price) but also receive subsidies, but this good will is hindered by the confusion of recycling channels. We have revised the related content to explain this problem on lines 96-99.

Point 8: Line 276 and following: "recycles" I'd recommend avoiding "recycles" when "collects" or "returns" is meant. "Recycling" has a very specific connotation that is not correct in this context.

Response 8: Thanks for pointing out the inappropriate using of "recycles" due to our carelessness. We have it revised by using “collects” or “collected” in such contexts. Thank you very much for your kindness.

Point 9: Line 283 and following: While I was reading through the models it wasn't clear to me if transportation costs are included.

Response 9: We apologize that our original statement on the costs in the model is not very clear, actually the transportation costs are not directly considered in the model, and can be understood to be included in the recovery costs, selling costs, etc. We have added the related description on line 286 to explain more clearly. Thank you very much for reminding us.

Point 10: Line 327: I have missed something in the model description, because why are all of the prices determined jointly by manufacturer and seller when not all of the prices apply to both enterprises?

Response 10: Thank you very much for this question, which we mentioned in the model assumption 2. Based on the actual characteristics of agricultural machinery CLSC in China, the manufacturers and sellers are typically part of the same enterprise, so decisions are made collectively in the models. We have added the related description to explain more clearly in the model development part of each model (line 298, line 337, line 385, line 435 and line 484).

Point 11: Figure 8: The x-axis range of this graph is much, much larger than the range of the data being graphed and discussed. I recommend re-drawing this figure so the data fills more of the available space.

Response 11: We apologize that the original figure 8 was not well graphed, as you pointed out, and it has been revised as clear as possible in this paper. Thank you very much for reminding us.

Point 12: All figures and related discussion: Virtually every figure is missing a y-axis label, which is needed even if the y-axis is described in the caption or in the accompanying text. All axis labels should include units, again even if the units are given elsewhere (which I do not believe they are). Currency units are especially important for a country-specific analysis like this one. The figure text (not the CLSC diagrams, but the graphs specifically) is also quite small throughout and in general the graphs could be more readable.

Response 12: Thanks for pointing out the deficiencies in the figures and related discussion. We apologize that the figures were not well presented in our original paper. Now we have supplemented the y-axis label and currency units to all the figures and also enlarged the figure text to make it more readable. In addition, since the currency units of the values are all the same, we omit the units in the related discussion for clarity. Thank you very much for your constructive suggestions and kind understanding that do support our work.

Point 13: Lines 549-551: This is a long, long list of parameters and numbers that have no units and aren't defined in the surrounding discussion. Instead of this list, I recommend a table with all parameters clearly defined and all units listed.

Response 13: We apologize that the assumed values for model variables and parameters were not well presented in the original Numerical Analysis section. According to your constructive suggestions, we have added the Table 10 on line 571 to clearly list all the assumed values for model variables and parameters and also supplemented the units to the values that have units. In addition, since the variables and parameters have been defined in Table 3 – Table 7, redefining these variables would make Table 10 very long, so we haven't redefined them here. Thank you very much for your kindness.

Point 14: Line 606: Are "opportunity costs" included explicitly in the model, or is the phrase being used more generally?

Response 14: The "opportunity costs" here is a general description. For clarity, the original statement has been replaced by “ensuring that all parties are willing to participate” (line 590 and line 622). Thank you very much for reminding us.

Point 15: Lines 679-680: This is a really exciting conclusion! I recommend emphasizing this finding much more heavily, as it's an insight that could be really impactful in the developing of CLSCs.

Response 15: It is very kind of you to provide us with valuable and insightful suggestions and comments, which are highly valuable in helping us improve the quality of this paper. And we have revised the related content to elaborate in more detail on this finding, as we mentioned in Response 2. Thank you very much for guiding us.

Point 16: In the beginning of the paper, there is much emphasis on the fact that this work focuses on Chinese CLSCs and how provinces in China are required to deal with end-of-life machinery. However, later on in the paper the results and conclusions are worded much more generally. I recommend the authors re-iterate the China-only nature of these insights - or, alternatively, state up-front that although the present work focuses on China, the resulting insights are more broadly applicable (also explain why).

Response 16: We really appreciate your valuable and insightful suggestions and comments, which are crucial in improving the quality of this paper, and should deserve our full respect. Thanks for pointing out the deficiencies in the discussion of the results and conclusions. Now we have revised the related content to discuss the managerial insights from the Chinese perspective, and present our recommendations for the Chinese government and the enterprises in agricultural machinery remanufacturing industry in Conclusions section based on the actual situation in China. And we have also added the statement that our conclusions can provide some references for other countries with similar experiences as China in agricultural machinery remanufacturing on lines 790-791.

Point 17: A note on transparency: The authors might consider publishing their calculations (whether as a codebase, spreadsheet, or other format), input datasets, and numerical results along with the article, provided there are no concerns about proprietary data. It's certainly not a requirement for the article to be published but it's good practice for math-heavy research like this.

Response 17: Thank you for your suggestion, which is of great importance for us to improve the quality of this paper. Actually, we calculate and plot the profits and other values of each model with different parameter settings through the software Matlab. And based on the results, we compare and obtain the optimal profit range of all parties in the supply chain under different transfer payments. Therefore, we have presented the Matlab code and dataset for applicable conditions of the SMM and SDM models as a representative example in the Supplementary Materials file. Thus, readers can calculate and reproduce the results in the tables and figures in Numerical Analysis section by adjusting some functions and parameter settings. Thank you very much for your kindness.

Reviewer 2 Report

The study examined five agricultural machinery recycling and remanufacturing channels and analyzed the pricing decisions of supply chain members according to different actors involved in recovery, dismantling, and remanufacturing. For that, the author analysed the applicable condition of each model and the influence of multiple factors. Since the closed-loop supply chain for scrapped agricultural machines remanufacturing is a crucial problem for promoting the development of agricultural mechanization, the article gives a great contribution to research and decision-makers. The article is well presented and contributes to a valuable debate in the discipline. No corrections need to be done.

Author Response

Response to Reviewer 2 Comments

Point 1: The study examined five agricultural machinery recycling and remanufacturing channels and analyzed the pricing decisions of supply chain members according to different actors involved in recovery, dismantling, and remanufacturing. For that, the author analysed the applicable condition of each model and the influence of multiple factors. Since the closed-loop supply chain for scrapped agricultural machines remanufacturing is a crucial problem for promoting the development of agricultural mechanization, the article gives a great contribution to research and decision-makers. The article is well presented and contributes to a valuable debate in the discipline. No corrections need to be done.

Response 1: Dear reviewer, firstly, thank you very much for your commendation, which is a very rewarding recognition to our efforts. According to the constructive suggestions from reviewers, we have carefully revised this paper to further improve the quality. The revisions to the manuscript have been marked up by using the “Track Changes” function of MS Word. Your acknowledgement will be highly appreciated. Thank you very much for your kindness.

Reviewer 3 Report

I have a couple of comments in order to advance the article:

Lines 28-31 please cite.

Please avoid to use the word “must”.

Lines 39-49 please cite. Citation is particularly important for proving two reasons for scrapping agricultural machinery.

Lines 50-51 please cite.

Generally, when you make a statement you should prove with the previous researches. There are a lot of pieces of the text where you need to provide the references. I will not list them for the rest of the paper.

Please provide sources for the tables and figures.

The aim of the study and research questions are not clearly defined in the introduction.

Please, try to make links between your results and previously published studies.

Also, try to strengthen your conclusion with discussed academic works.

Hereby, I suggest a couple of articles to include in your study:

1.       https://sciendo.com/es/article/10.2478/euco-2020-0014 : this article underlines the role of networking and collaborative approach of the stakeholders for reaching the goal and strengthen the industries. It might help you to better describe the role of governmental interventions (subsidies, regulations, taxes, etc.) for expand the experience of remanufacturing agricultural machineries.

2.       https://www.inderscienceonline.com/doi/abs/10.1504/JGBA.2020.110612 : this article highlights the importance of collaboration between companies, sharing knowledge and experience in order to strengthen the field.

Overall merit: the article is interesting, high quality original research. It has a great potential for the contribution for theory and practice. The major problem is the citation. Authors need to strengthen their statements with the previously published literature. Now they are telling a story with concrete facts without proving their words with academic literature.

Author Response

Response to Reviewer 3 Comments

Point 1: Lines 28-31 please cite.

Response 1: We do apologize for the lack of reference in our original paper, and it has been revised to strengthen the statements with the previously published literature. We have supplemented the relevant reference on line 31 in the revised paper. Thank you very much.

Point 2: Please avoid to use the word “must”.

Response 2: Thank you very much for pointing out the inappropriate using of "must" due to our carelessness, and they have been replaced by “need to” or “should” in this paper.

Point 3: Lines 39-49 please cite. Citation is particularly important for proving two reasons for scrapping agricultural machinery.

Response 3: Thank you very much for your kind suggestion, and we have supplemented the relevant references on line 41 and line 47 in the revised paper.

Point 4: Lines 50-51 please cite.

Response 4: Thank you very much for reminding us, and we have supplemented the relevant reference on line 51 in the revised paper.

Point 5: Generally, when you make a statement you should prove with the previous researches. There are a lot of pieces of the text where you need to provide the references. I will not list them for the rest of the paper.

Response 5: We do apologize for the lack of reference in our original paper, and thank you for your suggestion, which is of great importance for us to improve the quality of this paper. Now it has been revised and enhanced as specific as possible to strengthen the statements with the previously published literature. And due to some facts about the recycling and remanufacturing of agricultural machinery in China are mainly published in Chinese journals, there are three of the references were written in Chinese. Thank you very much for your kindness.

Point 6: Please provide sources for the tables and figures.

Response 6: We apologize that the sources for the tables and figures were not very clear in our original paper. Now we have supplemented the source of Table 1 on lines 74-75. Furthermore, the Proof of Proposition 1 and Proof of Proposition 2 have been revised and enhanced as the representative proofs in Appendix A. These can provide the sources for the results of derivation in Model Development section which were summarized in Table 8 and Table 9. In addition, as a representative example, the Matlab code and dataset for applicable conditions of the SMM and SDM models are presented in the Supplementary Materials file. Thus, readers can calculate and reproduce the results in the tables and figures in Numerical Analysis section by adjusting some functions and parameter settings. And referring to other published papers in Sustainability, we omit the source of Table 2-11 and Figure 1-11 since they are based on our processing of modeling, calculations and simulation. Thank you very much for your constructive suggestions and kind understanding that do support our work.

Point 7: The aim of the study and research questions are not clearly defined in the introduction.

Response 7: We apologize that our original statement on the aim of the study and research questions in Introduction section are not very clear, and it has been revised to explain the aim of this paper as specific as possible on lines 101-106. Thank you very much for reminding us.

Point 8: Please, try to make links between your results and previously published studies.

Also, try to strengthen your conclusion with discussed academic works.

Hereby, I suggest a couple of articles to include in your study:

  1. https://sciendo.com/es/article/10.2478/euco-2020-0014 : this article underlines the role of networking and collaborative approach of the stakeholders for reaching the goal and strengthen the industries. It might help you to better describe the role of governmental interventions (subsidies, regulations, taxes, etc.) for expand the experience of remanufacturing agricultural machineries.
  2. https://www.inderscienceonline.com/doi/abs/10.1504/JGBA.2020.110612 : this article highlights the importance of collaboration between companies, sharing knowledge and experience in order to strengthen the field.

Response 8: It is very kind of you to provide us with valuable and insightful suggestions and comments, which are highly valuable in helping us improve the quality of this paper, and should deserve our full respect. According to your constructive suggestions, we have revised and enhanced the related content to strengthen the discussion of results and conclusion with previously academic works. Firstly, we focused on the benefits of supply chain collaboration and explored the possible causes in combination with the literature. To some extent, it is similar to the value co-creation strategy in logistics industry, collaboration between the core enterprises in agricultural machinery CLSC and the third-party enterprises can mediate the relationship between resource commitment and performance. Secondly, as you recommended, we presented our recommendations for the Chinese government and the enterprises in agricultural machinery remanufacturing industry. Specifically, for new entrants in such industry, more open and inclusive channels should be chosen rather than those that exclude third-party enterprises. In addition, the agricultural machinery recycling and remanufacturing industry in China is mainly composed of small and medium-sized enterprises. In order to maintain and strengthen supply chain cooperation scenarios, the Community of Practice can be established to create strong network among supply chain members and eliminate the informative gap. And relying on the government's strong policy enforcement, the Chinese government should introduce effective policies that encourage collaboration and infor-mation/knowledge exchange between key actors to promote the agricultural machinery recycling and remanufacturing. Further, we discussed the government policies when the supply of remanufactured products exceeded demand, and suggested that some targeted policies can be created in China including recovery price subsidies, tax exemptions and funding research activities based on relevant literature. The above content has been supplemented to the discussion of results (lines 697-700) and conclusions (lines 824-850). Thank you very much for guiding us.

Point 9: Overall merit: the article is interesting, high quality original research. It has a great potential for the contribution for theory and practice. The major problem is the citation. Authors need to strengthen their statements with the previously published literature. Now they are telling a story with concrete facts without proving their words with academic literature.

Response 9: Dear reviewer, firstly, thank you very much for your commendation, which is a very rewarding recognition to our efforts. We are particularly grateful for your valuable and insightful suggestions and comments, which are very important for us to improve the quality of this paper, and should deserve our full respect. According to the constructive suggestions from you and other reviewers, we have devoted a great deal of time and effort to revise this paper. The Literature review section and Conclusions section have been greatly improved, and we have supplemented the references, proofs of derivation process and the example of Matlab code and dataset to further enhance the quality of this paper. And we have also carefully edited the manuscript to eliminate other shortcomings. The revisions to the manuscript have been marked up by using the “Track Changes” function of MS Word. We appreciate all the comments and suggestions, and hope that our revisions match all the requirements. If you have any question about this paper, please don’t hesitate to let me know. Your acknowledgement will be highly appreciated. We will appreciate to receive further comments and suggestions about this paper and actively make the revision in order to enhance the quality of this paper.

Round 2

Reviewer 1 Report

I appreciate the authors' attention to detail in this revision. With the addition of the supporting datasets and code and the rewriting of the literature review section, my major concerns from the first round of revisions have been addressed. I think some light editing just for grammar would be beneficial before publication, but other than that I have no further changes to suggest.

Reviewer 3 Report

Authors have considerably improved the paper and they took into consideration my previous comments. I suggest to publish the paper in the present form.